# Learning advanced mathematical computations from examples

**François Charton**[*]
Facebook AI Research
fcharton@fb.com

**Amaury Hayat**[*]
Ecole des Ponts Paristech,
Rutgers University - Camden
amaury.hayat@enpc.fr

**Guillaume Lample**
Facebook AI Research
glample@fb.com

## Abstract

Using transformers over large generated datasets, we train models to learn mathematical properties of differential systems, such as local stability, behavior at infinity and controllability. We achieve near perfect prediction of qualitative characteristics, and good approximations of numerical features of the system. This demonstrates that neural networks can learn to perform complex computations, grounded in advanced theory, from examples, without built-in mathematical knowledge.

## 1 Introduction

Scientists solve problems of mathematics by applying rules and computational methods to the data at hand. These rules are derived from theory, they are taught in schools or implemented in software libraries, and guarantee that a correct solution will be found. Over time, mathematicians have developed a rich set of computational tools that can be applied to many problems, and have been said to be "unreasonably effective" (Wigner, 1960).

Deep learning, on the other hand, learns from examples and solves problems by improving a random initial solution, without relying on domain-related theory and computational rules. Deep networks have proven to be extremely efficient for a large number of tasks, but struggle on relatively simple, rule-driven arithmetic problems (Saxton et al., 2019; Trask et al., 2018; Zaremba and Sutskever, 2014).

Yet, recent studies show that deep learning models can learn complex rules from examples. In natural language processing, models learn to output grammatically correct sentences without prior knowledge of grammar and syntax (Radford et al., 2019), or to automatically map one language into another (Bahdanau et al., 2014; Sutskever et al., 2014). In mathematics, deep learning models have been trained to perform logical inference (Evans et al., 2018), SAT solving (Selsam et al., 2018) or basic arithmetic (Kaiser and Sutskever, 2015). Lample and Charton (2020) showed that transformers can be trained from generated data to perform symbol manipulation tasks, such as function integration and finding formal solutions of ordinary differential equations.

In this paper, we investigate the use of deep learning models for complex mathematical tasks involving both symbolic and numerical computations. We show that models can predict the qualitative and quantitative properties of mathematical objects, without built-in mathematical knowledge. We consider three advanced problems of mathematics: the local stability and controllability of differential systems, and the existence and behavior at infinity of solutions of partial differential equations. All three problems have been widely researched and have many applications outside of pure mathematics. They have known solutions that rely on advanced symbolic and computational techniques, from formal differentiation, Fourier transform, algebraic full-rank conditions, to function evaluation, matrix inversion, and computation of complex eigenvalues. We find that neural networks can solve these problems with a very high accuracy, by simply looking at instances of problems and their solutions, while being totally unaware of the underlying theory. In one of the quantitative problems

---

[*] Equal contribution, names in alphabetic order.

where several solutions are possible (predicting control feedback matrix), neural networks are even able to predict different solutions that those generated with the mathematical algorithms we used for training.

After reviewing prior applications of deep learning to related areas we introduce the three problems we consider, describe how we generate datasets, and detail how we train our models. Finally, we present our experiments and discuss their results.

## 2 Related work

Applications of neural networks to differential equations have mainly focused on two themes: numerical approximation and formal resolution. Whereas most differential systems and partial differential equations cannot be solved explicitly, their solutions can be approximated numerically, and neural networks have been used for this purpose (Lagaris et al., 1998; 2000; Lee and Kang, 1990; Rudd, 2013; Sirignano and Spiliopoulos, 2018). This approach relies on the universal approximation theorem, that states that any continuous function can be approximated by a neural network with one hidden layer over a wide range of activation functions (Cybenko, 1989; Hornik et al., 1990; Hornik, 1991; Petersen and Voigtlaender, 2018; Pinkus, 1999). This has proven to be especially efficient for high dimensional problems.

For formal resolution, Lample and Charton (2020) proposed several approaches to generate arbitrarily large datasets of functions with their integrals, and ordinary differential equations with their solutions. They found that a transformer model (Vaswani et al., 2017) trained on millions of examples could outperform state-of-the-art symbolic frameworks such as Mathematica or MATLAB (Wolfram-Research, 2019; MathWorks, 2019) on a particular subset of equations. Their model was used to guess solutions, while verification (arguably a simpler task) was left to a symbolic framework (Meurer et al., 2017). Arabshahi et al. (2018a;b) proposed to use neural networks to verify the solutions of differential equations, and found that Tree-LSTMs (Tai et al., 2015) were better than sequential LSTMs (Hochreiter and Schmidhuber, 1997) at generalizing beyond the training distribution.

Other approaches investigated the capacity of neural networks to perform arithmetic operations (Kaiser and Sutskever, 2015; Saxton et al., 2019; Trask et al., 2018) or to run short computer programs (Zaremba and Sutskever, 2014). More recently, Saxton et al. (2019) found that neural networks were good at solving arithmetic problems or at performing operations such as differentiation or polynomial expansion, but struggled on tasks like prime number decomposition or on primality tests that require a significant number of steps to compute. Unlike the questions considered here, most of those problems can be solved by simple algorithmic computations.

## 3 Differential systems and their stability

A differential system of degree $n$ is a system of $n$ equations of $n$ variables $x_1(t), ..., x_n(t)$,

$$\frac{dx_i(t)}{dt} = f_i\big(x_1(t), x_2(t), ..., x_n(t)\big), \qquad \text{for} \quad i = 1...n$$

or, in vector form, with $x \in \mathbb{R}^n$ and $f : \mathbb{R}^n \to \mathbb{R}^n$,

$$\frac{dx(t)}{dt} = f\big(x(t)\big)$$

Many problems can be set as differential systems. Special cases include n-th order ordinary differential equations (letting $x_1 = y$, $x_2 = y'$, ... $x_n = y^{(n-1)}$), systems of coupled differential equations, and some particular partial differential equations (separable equations or equations with characteristics). Differential systems are one of the most studied areas of mathematical sciences. They are found in physics, mechanics, chemistry, biology, and economics as well as in pure mathematics. Most differential systems have no explicit solution. Therefore, mathematicians have studied the properties of their solutions, and first and foremost their stability, a notion of paramount importance in many engineering applications.

## 3.1 Local stability

Let $x_e \in \mathbb{R}^n$ be an equilibrium point, that is, $f(x_e) = 0$. If all solutions $x(t)$ converge to $x_e$ when their initial positions $x(0)$ at $t = 0$ are close enough, the equilibrium is said to be locally stable (see Appendix B for a proper mathematical definition). This problem is well known, if $f$ is differentiable in $x_e$, an answer is provided by the Spectral Mapping Theorem (SMT) (Coron, 2007, Theorem 10.10):

**Theorem 3.1.** *Let $J(f)(x_e)$ be the Jacobian matrix of $f$ in $x_e$ (the matrix of its partial derivatives relative to its variables). Let $\lambda$ be the largest real part of its eigenvalues. If $\lambda$ is positive, $x_e$ is an unstable equilibrium. If $\lambda$ is negative, then $x_e$ is a locally stable equilibrium.*

Predicting the stability of a given system at a point $x_e$ is our first problem. We will also predict $\lambda$, which represents the speed of convergence when negative, in a second experiment. Therefore, to apply the SMT, we need to:

1. differentiate each function with respect to each variable, obtain the formal Jacobian $J(x)$

$$f(x) = \begin{pmatrix} \cos(x_2) - 1 - \sin(x_1) \\ x_1^2 - \sqrt{1 + x_2} \end{pmatrix}, \quad J(x) = \begin{pmatrix} -\cos(x_1) & -\sin(x_2) \\ 2x_1 & -(2\sqrt{1 + x_2})^{-1} \end{pmatrix}$$

2. evaluate $J(x_e)$, the Jacobian in $x_e$ (a real or complex matrix)

$$x_e = (0.1, ...0.1) \in \mathbb{R}^n, \quad J(x_e) = \begin{pmatrix} -\cos(0.1) & -\sin(0.1) \\ 0.2 & -(2\sqrt{1 + 0.1})^{-1} \end{pmatrix},$$

3. calculate the eigenvalues $\lambda_i, i = 1...n$ of $J(x_e)$

$$\lambda_1 = -1.031, \quad \lambda_2 = -0.441$$

4. compute $\lambda = -\max(\text{Real}(\lambda_i))$ and return the stability (resp. $\lambda$ the speed of convergence)

$$\lambda = 0.441 > 0 \rightarrow \text{locally stable with decay rate } 0.441$$

## 3.2 Control theory

One of the lessons of the spectral mapping theorem is that instability is very common. In fact, unstable systems are plenty in nature (Lagrange points, epidemics, satellite orbits, etc.), and the idea of trying to control them through external variables comes naturally. This is the controllability problem. It has a lot of practical applications, including space launch and the landing on the moon, the US Navy automated pilot, or recently autonomous cars (Bernhard et al., 2017; Minorsky, 1930; Funke et al., 2016). Formally, we are given a system

$$\frac{dx}{dt} = f\big(x(t), u(t)\big), \tag{1}$$

where $x \in \mathbb{R}^n$ is the state of the system. We want to find a function $u(t) \in \mathbb{R}^p$, the control action, such that, beginning from a position $x_0$ at $t = 0$, we can reach a position $x_1$ at $t = T$ (see Appendix B). The first rigorous mathematical analysis of this problem was given by Maxwell (1868), but a turning point was reached in 1963, when Kalman gave a precise condition for a linear system (Kalman et al., 1963), later adapted to nonlinear system:

**Theorem 3.2** (Kalman condition). *Let $A = \partial_x f(x_e, u_e)$ and $B = \partial_u f(x_e, u_e)$, if*

$$Span\{A^i Bu : u \in \mathbb{R}^m, i \in \{0, ..., n - 1\}\} = \mathbb{R}^n, \tag{2}$$

*then the system is locally controllable around $x = x_e, u = u_e$.*

When this condition holds, a solution to the control problem that makes the system locally stable in $x_e$ is $u(t) = u_e + K(x(t) - x_e)$ (c.f. Coron (2007); Kleinman (1970); Lukes (1968)

and appendix B.4 for key steps of the proof), where $K$ is the $m \times n$ control feedback matrix:

$$K = -B^{tr} \left( e^{-AT} \left[ \int_0^T e^{-At} BB^{tr} e^{-A^{tr}t} dt \right] e^{-A^{tr}T} \right)^{-1}. \tag{3}$$

In the non-autonomous case, where $f = f(x, u, t)$ (and $A$ and $B$) depends on $t$, (2) can be replaced by:

$$\text{Span}\{D_i u : u \in \mathbb{R}^m, i \in \{0, ..., 2n-1\} = \mathbb{R}^n\}, \tag{4}$$

where $D_0(t) = B(t)$ and $D_{i+1}(t) = D_i'(t) - A(t)D_i(t)$. All these theorems make use of advanced mathematical results, such as the Cayley-Hamilton theorem, or LaSalle invariance principle. Learning them by predicting controllability and computing the control feedback matrix $K$ is our second problem. To measure whether the system is controllable at a point $x_e$, we need to:

1. differentiate the system with respect to its internal variables, obtain $A(x, u)$
2. differentiate the system with respect to its control variables, obtain $B(x, u)$
3. evaluate $A$ and $B$ in $(x_e, u_e)$
4. calculate the controllability matrix $C$ with (2) (resp. (4) if non-autonomous)
5. calculate the rank $d$ of $C$, if $d = n$, the system is controllable
6. (optionally) if $d = n$, compute the control feedback matrix $K$ with (3)

In: $f(x, u) = \begin{pmatrix} \sin(x_1^2) + \log(1 + x2) + \frac{\text{atan}(ux_1)}{1+x_2} \\ x_2 - e^{x_1 x_2} \end{pmatrix}, \begin{matrix} x_e = [0.1] \\ u_e = 1 \end{matrix},$ Out: $\begin{cases} n - d = 0 \\ \text{System is controllable} \\ K = (-22.8 \quad 44.0) \end{cases}$

A step by step derivation of this example is given in Section A of the appendix.

### 3.3 Stability of partial differential equations using Fourier Transform

Partial Differential Equations (PDEs) naturally appear when studying continuous phenomena (e.g. sound, electromagnetism, gravitation). Over such problems, ordinary differential systems are not sufficient. Like differential systems, PDEs seldom have explicit solutions, and studying their stability has many practical applications. It is also a much more difficult subject, where few general theorems exist. We consider linear PDEs of the form

$$\partial_t u(t, x) + \sum_{|\alpha| \leq k} a_\alpha \partial_x^\alpha u(t, x) = 0, \tag{5}$$

where $t$, $x \in \mathbb{R}^n$, and $u(t, x)$ are time, position, and state. $\alpha = (\alpha_1, ..., \alpha_n) \in \mathbb{R}^n$ is a multi-index and $a_\alpha$ are constants. Famous examples of such problems include the heat equation, transport equations or Schrodinger equation (Evans, 2010). We want to determine whether a solution $u(t, x)$ of (5) exists for a given an initial condition $u(0, x) = u_0$, and if it tends to zero as $t \to +\infty$. This is mathematically answered (see appendix B.4 and Evans (2010); Bahouri et al. (2011) for similar arguments) by:

**Proposition 3.1.** *Given $u_0 \in \mathcal{S}'(\mathbb{R}^n)$, the space of tempered distribution, there exists a solution $u \in \mathcal{S}'(\mathbb{R}^n)$ if there exists a constant $C$ such that*

$$\forall \xi \in \mathbb{R}^n , \ \widetilde{u}_0(\xi) = 0 \ \text{ or } \ \text{Real}(f(\xi)) > C, \tag{6}$$

*where $\widetilde{u}_0$ is the Fourier transform of $u_0$ and $f(\xi)$ is the Fourier polynomial associated with the differential operator $D_x = \sum_{|\alpha| \leq k} a_\alpha \partial_x^\alpha$. In addition, if $C > 0$, this solution $u(t, x)$ goes to zero when $t \to +\infty$.*

Learning this proposition and predicting, given an input $D_x$ and $u_0$, whether a solution $u$ exists, if so, whether it vanishes at infinite time, will be our third and last problem.

To predict whether our PDE has a solution under given initial conditions, and determine its behavior at infinity, we need to: find the Fourier polynomial $f(\xi)$ associated to $D_x$; find the Fourier transform $\tilde{u}_0(\xi)$ of $u_0$; minimize $f(\xi)$ on $\mathcal{F}$; output $(0,0)$ if this minimum is infinite, $(1,0)$ is finite and negative, $(1,1)$ if finite and positive. Optionally, output $\mathcal{F}$. A step by step example is given in Appendix A.

$$\text{In: } D_x = 2\partial_{x_0}^2 + 0.5\partial_{x_1}^2 + \partial_{x_2}^4 - 7\partial_{x_0,x_1}^2 - 1.5\partial_{x_1}\partial_{x_2}^2,$$
$$\text{Out:}(1, 0) \;\rightarrow\; \text{there exists a solution } u \text{ ; it does not vanish at } t \to +\infty$$

## 4  DATASETS AND MODELS

To generate datasets, we randomly sample problems and compute their solutions with mathematical software (Virtanen et al., 2020; Meurer et al., 2017) using the techniques described in Section 3. For stability and controllability, we generate differential systems with $n$ equations and $n + q$ variables (i.e. $n$ random functions, $q > 0$ for controllability).

Following Lample and Charton (2020), we generate random functions by sampling unary-binary trees, and randomly selecting operators, variables and integers for their internal nodes and leaves. We use $+, -, \times, /, \exp, \log, \text{sqrt}, \sin, \cos, \tan, \sin^{-1}, \cos^{-1}, \tan^{-1}$ as operators, and integers between $-10$ and $10$ as leaves. When generating functions with $n + q$ variables, we build trees with up to $2(n + q + 1)$ operators.

Generated trees are enumerated in prefix order (normal Polish notation) and converted into sequences of tokens compatible with our models. Integers and floating point reals are also represented as sequences: 142 as [INT+, 1, 4, 2], and 0.314 as [FLOAT+, 3, DOT, 1, 4, E, INT-, 1]. A derivation of the size of the problem space is provided in appendix D.4.

**Local stability**  Datasets for local stability include systems with 2 to 6 equations (in equal proportion). Functions that are not differentiable at the equilibrium $x_e$ and degenerate systems are discarded. Since many of the operators we use are undefined at zero, setting $x_e = 0$ would result in biasing the dataset by reducing the frequency of operators like division, square root, or logarithms. Instead, we select $x_e$ with all coordinates equal to 0.01 (denoted as $x_e = [0.01]$). This is, of course, strictly equivalent mathematically to sampling systems with equilibrium at the origin or at any other point.

When predicting overall stability, since stable systems become exponentially rare as dimension increases, we use rejection sampling to build a balanced dataset with 50% stable systems. When predicting convergence speed, we work from a uniform (i.e. unbalanced) sample. The value of $\lambda$ at $x_e$ is expressed as a floating point decimal rounded to 4 significant digits. For this problem, we generate two datasets with over 50 million systems each.

**Control theory**  Datasets for automonous control include systems with 3 to 6 equations, and 4 to 9 variables (1 to 3 control variables). In the non-autonomous case, we generate systems with 2 or 3 equations. As above, we discard undefined or degenerate systems. We also skip functions with complex Jacobians in $x_e$ (since the Jacobian represents local acceleration, one expects its coordinates to be real). We have $x_e = [0.5]$ or $[0.9]$.

In the autonomous case, more than 95% of the systems are controllable. When predicting controllability, we use rejection sampling to create a balanced dataset. In the non-autonomous case, we use a uniform sample with 83% controllable cases. Finally, to predict feedback matrices, we restrict generation to controllable systems and express the matrix as a sequence of floating point decimals. All 3 datasets have more than 50 million examples each.

**Stability of partial differential equations using Fourier Transform**  We generate a differential operator (a polynomial in $\partial_{x_i}$) and an initial condition $u_0$. $u_0$ is the product of $n$ functions $f(a_j x_j)$ with known Fourier transforms, and $d$ operators $\exp(ib_k x_k)$, with $0 \leq d \leq 2n$ and $a_j, b_k \in \{-100, \ldots, 100\}$. We calculate the existence of solutions, their behavior when $t \to +\infty$, and the set of frequencies, and express these three values as a sequence of 2 Booleans and floating point decimals. Our dataset is over 50 million examples.

**Models and evaluation**    In all experiments, we use a transformer architecture with 8 attention heads. We vary the dimension from 64 to 1024, and the number of layers from 1 to 8. We train our models with the Adam optimizer (Kingma and Ba, 2014), a learning rate of $10^{-4}$ and the learning rate scheduler in Vaswani et al. (2017), over mini-batches of 1024 examples. Additional information can be found in appendix D.1. Training is performed on 8 V100 GPUs with float16 operations. Our qualitative models (predicting stability, controllability and existence of solutions) were trained for about 12 hours, but accuracies close to the optimal values were reached after about 6 hours. Learning curves for this problem can be found in appendix D.3. On quantitative models, more training time and examples were needed: 76 hours for convergence speed, 73 hours for control matrices.

Evaluation is performed on held-out validation and test sets of 10000 examples. We ensure that validation and test examples are never seen during training (given the size of the problem space, this never happens in practice). Model output is evaluated either by comparing it with the reference solution or using a problem-specific metric.

## 5   EXPERIMENTS

### 5.1   PREDICTING QUALITATIVE PROPERTIES OF DIFFERENTIAL SYSTEMS

In these experiments, the model is given $n$ functions $f : \mathbb{R}^{n+p} \to \mathbb{R}$ ($n \in \{2, \dots, 6\}$, $p = 0$ for stability, $p > 0$ for controllability) and is trained to predict whether the corresponding system is stable, resp. controllable, at a given point $x_e$. This is a classification problem.

To provide a baseline for our results, we use fastText (Joulin et al., 2016), a state-of-the-art text classification tool, which estimates, using a bag of words model, the probability of a qualitative feature (stability) conditional to the distribution of tokens and of small fixed sequences (N-grams of up to five tokens). Such a model can detect simple correlations between inputs and outputs, such as the impact on stability of the presence of a given operator, or the number of equations in the system. It would also find out obvious solutions, due to the specifics of one problem or glitches in the data generator. FastText was trained over 2 million examples from our dataset (training over larger sets does not improve accuracy).

A 6-layer transformer with 512 dimensions correctly predicts the system stability in 96.4% of the cases. Since the dataset is balanced, random guessing would achieve 50%. FastText achieves 60.6%, demonstrating that whereas some easy cases can be learnt by simple text classifiers, no trivial general solution exists for this dataset. Prediction accuracy decreases with the degree, but remains high even for large systems (Table 1).

Table 1: **Accuracy of predictions of stability (chance level: 50%)**

|  | Degree 2 | Degree 3 | Degree 4 | Degree 5 | Overall | FastText |
|---|---|---|---|---|---|---|
| Accuracy | 98.2 | 97.3 | 95.9 | 94.1 | 96.4 | 60.6 |

For autonomous controllability over a balanced dataset, a 6-layer transformer with 512 dimensions correctly predicts 97.4% of the cases. The FastText baseline is 70.5%, above the 50% chance level. Whereas accuracy increases with model size (dimension and number of layers), even very small models (dimension 64 and only 1 or 2 layers) achieve performance over 80%, above the FastText baseline (Table 2).

Table 2: **Accuracy of autonomous control task over a balanced sample of systems with 3 to 6 equations.**

|  | Dimension 64 | Dimension 128 | Dimension 256 | Dimension 512 | FastText |
|---|---|---|---|---|---|
| 1 layers | 81.0 | 85.5 | 88.3 | 90.4 | - |
| 2 layers | 82.7 | 88.0 | 93.9 | 95.5 | - |
| 4 layers | 84.1 | 89.2 | 95.6 | 96.9 | - |
| 6 layers | 84.2 | 90.7 | 96.3 | **97.4** | **70.5** |

For non-autonomous systems, our dataset features systems of degree 2 and 3, 83% controllable. FastText achieves 85.3%, barely above the chance level of 83%. This shows that text classifiers have difficulty handling difficult problems like this one, even in low dimensions. Our model achieves 99.7% accuracy. Again, small models, that would be unsuitable for natural language processing, achieve near perfect accuracy (Table 3).

Table 3: **Accuracy for non-autonomous control over systems with 2 to 3 equations.**

|  | Dimension 64 | Dimension 128 | Dimension 256 | Dimension 512 | FastText |
|---|---|---|---|---|---|
| 1 layer | 97.9 | 98.3 | 98.5 | 98.9 | - |
| 2 layers | 98.4 | 98.9 | 99.3 | 99.5 | - |
| 4 layers | 98.6 | 99.1 | 99.4 | 99.6 | - |
| 6 layers | 98.7 | 99.1 | 99.5 | **99.7** | **85.3** |

## 5.2 Predicting numerical properties of differential systems

**Speed of convergence**  In these experiments, the model is trained to predict $\lambda$, the convergence speed to the equilibrium, up to a certain precision. Here, we consider predictions to be correct when they fall within 10% of the ground truth. Further experiments with different levels of precision (2, 3 or 4 decimal digits) are provided in Appendix C.

A model with 8 layers and a dimension of 1024 predicts convergence speed with an accuracy of 86.6% overall. While reasonably good results can be achieved with smaller models, the accuracy decrease quickly when model size falls under a certain value, unlike when qualitative properties were predicted. Table 4 summarizes the results.

Table 4: **Prediction of local convergence speed (within 10%).**

|  | Degree 2 | Degree 3 | Degree 4 | Degree 5 | Degree 6 | Overall |
|---|---|---|---|---|---|---|
| 4 layers, dim 512 | 88.0 | 74.3 | 63.8 | 54.2 | 45.0 | 65.1 |
| 6 layers, dim 512 | 93.6 | 85.5 | 77.4 | 71.5 | 64.9 | 78.6 |
| 8 layers, dim 512 | 95.3 | 88.4 | 83.4 | 79.2 | 72.4 | 83.8 |
| 4 layers, dim 1024 | 91.2 | 80.1 | 71.6 | 61.8 | 54.4 | 71.9 |
| 6 layers, dim 1024 | 95.7 | 89.0 | 83.4 | 78.4 | 72.6 | 83.8 |
| 8 layers, dim 1024 | **96.3** | **90.4** | **86.2** | **82.7** | **77.3** | **86.6** |

**Control feedback matrices**  In these experiments, we train the model (6 layers, 512 dimensions) to predict a feedback matrix ensuring stability of an autonomous system. We use two metrics to evaluate accuracy:

1) prediction within 10% of all coefficients in the target matrix $K$ given by (3) and provided in the training set,

2) verifying that the model outputs a correct feedback matrix $K_1$, i.e. that all eigenvalues in $A + BK_1$ have negative real parts. This makes more mathematical sense, as it verifies that the model provides an actual solution to the control problem (like a differential equation, a feedback control problem can have many different solutions).

Using the first metric, 15.8% of target matrices $K$ are predicted with less than 10% error. Accuracy is 50.0% for systems with 3 equations, but drops fast as systems becomes larger. These results are very low, although well above chance level ($<0.0001\%$). With the second metric (i.e. the one that actually matters mathematically), we achieve 66.5% accuracy, a much better result. Accuracy decreases with system size, but even degree 6 systems, with $1 \times 6$ to $3 \times 6$ feedback matrices, are correctly predicted 41.5% of the time. Therefore, while the model fails to approximate $K$ to a satisfactory level, it does learn to predict correct solutions to the control problem in 66.5% of the cases. This result is very surprising, as it suggests that a mathematical property characterizing feedback matrices might have been learned.

Table 5: **Prediction of feedback matrices - Approximation vs. correct mathematical feedback.**

|  | Degree 3 | Degree 4 | Degree 5 | Degree 6 | Overall |
|---|---|---|---|---|---|
| Prediction within 10% | 50.0 | 9.3 | 2.1 | 0.4 | 15.8 |
| Correct feedback matrix | 87.5 | 77.4 | 58.0 | 41.5 | 66.5 |

### 5.3 PREDICTING QUALITATIVE PROPERTIES OF PDES

In this setting, the model is given a differential operator $D_x$ and an initial condition $u_0$. It is trained to predict if a solution to $\partial_t u + D_x u = 0$ exists and, if so, whether it converges to 0 when $t \to +\infty$. The space dimension (i.e. dimension of $x$) is between 2 and 6.

In a first series of experiments the model is only trained to predict the existence and convergence of solutions. Overall accuracy is 98.4%. In a second series, we introduce an auxiliary task by adding to the output the frequency bounds $\mathcal{F}$ of $u_0$. We observe it significantly contributes to the stability of the model with respect to hyper-parameters. In particular, without the auxiliary task, the model is very sensitive to the learning rate scheduling and often fails to converge to something better than random guessing. However, in case of convergence, the model reaches the same overall accuracy, with and without auxiliary task. Table 6 details the results.

Table 6: **Accuracy on the existence and behavior of solutions at infinity.**

| Space dimension for $x$ | Dim 2 | Dim 3 | Dim 4 | Dim 5 | Dim 6 | Overall |
|---|---|---|---|---|---|---|
| Accuracy | 99.4 | 98.9 | 98.7 | 98.0 | 96.9 | 98.4 |

## 6 DISCUSSION

We studied five problems of advanced mathematics from widely researched areas of mathematical analysis. In three of them, we predict qualitative and theoretical features of differential systems. In two, we perform numerical computations. According to mathematical theory, solving these problems requires a combination of advanced techniques, symbolic and numerical, that seem unlikely to be learnable from examples. Yet, our model achieves more than 95% accuracy on all qualitative tasks, and between 65 and 85% on numerical computations.

When working from synthetic data, a question naturally arises about the impact of data generation on the results of experiments. In particular, one might wonder whether the model is exploiting a defect in the generator or a trivial property of the problems that allows for easier solutions. We believe this is very unlikely. First, because our results are consistent over different problems, using datasets generated with different techniques. Second, because a trivial solution would be found by the bag of words model we use as a baseline. And finally, because we build our datasets by direcly sampling problems from a distribution that includes all possible functions (up to the basis operators and the random number generator). This eliminates the biases that can result from sampling special instances or solutions (Yehuda et al., 2020). It also means that the training set is an extremely tiny sample of the whole problem space (over the 50 million examples generated, we did not get a single duplicate).

Learning from very large samples often raises questions about overfitting and generalisation out of the training distribution. Due to the size of the problem space, it is very unlikely that the model could memorize a large number of cases and interpolate between them. Note that because the space of functions from $\mathbb{R}^n$ to $\mathbb{R}^n$ has infinite dimension, the universal approximation theorem does not apply here. Note also that for some of our problems (e.g. local stability), mathematical theory states that solutions cannot be obtained by simple interpolation. To investigate out-of-distribution generalization, we modified our data generator to produce 10 new test sets for stability prediction. We changed the distribution of operators and variables, and experimented with systems with longer expressions and more equations. Table 7 (see Appendix C.2 for a detailed analysis) summarizes our key results.

Changes in the distribution of operators and variables have very little impact on accuracy, demonstrating that the model can generalize out of the training distribution. Our trained model also performs well on systems with longer expressions than the training data. This is interesting because generalizing to longer sequences is a known limitation of many sequence to sequence architectures. Finally, a model trained on systems with 2 to 5 equations predicts the stability of systems of 6 equations to high accuracy (78%). Being able to generalize to a larger problem space, with one additional variable, is a very surprising result, that tends to confirm that some mathematical properties of differential systems have been learned.

Table 7: **End to end stability: generalization over different test sets.**

|  | Overall | Degree 2 | Degree 3 | Degree 4 | Degree 5 |
|---|---|---|---|---|---|
| Baseline: training distribution | 96.4 | 98.4 | 97.3 | 95.9 | 94.1 |
| No trig operators | 95.7 | 98.8 | 97.3 | 95.5 | 91.2 |
| Variables and integers: 10% integers | 96.1 | 98.6 | 97.3 | 94.7 | 93.8 |
| Expression lengths: n+3 to 3n+3 | 89.5 | 96.5 | 92.6 | 90.0 | 77.9 |
| System degree: degree 6 | 78.7 |  |  |  |  |

It seems unlikely that the model follows the same mathematical procedure as human solvers. For instance, problems involving more computational steps, such as non-autonomous controllability, do not result in lower accuracy. Also, providing at train time intermediate results that would help a human calculator (frequencies for PDE, or Jacobians for stability) does not improve performance. Understanding how the model finds solutions would be very interesting, as no simpler solutions than the classical mathematical steps are known.

To this effect, we tried to analyze model behavior by looking at the attention heads and the tokens the models focus on when it predicts a specific sequence (following Clark et al. (2019)). Unfortunately, we were not able to extract specific patterns, and found that each head in the model, from the first layer onwards, attends many more tokens than in usual natural language tasks (i.e. attention weights tend to be uniformly distributed). This makes interpretation very difficult.

These results open many perspectives for transformers in fields that need both symbolic and numerical computations. There is even hope that our models could help solve mathematical problems that are still open. On a more practical level, they sometimes provide fast alternatives to classical solvers. The algorithmic complexity of transformer inference and classical algorithms for the problems we consider here is discussed in appendix E.1. However, for the problems in our dataset, the simpler and parallelizable computations used by transformers allow for 10 to 100 times shorter evaluation times (see Appendix E.2).

## 7 Conclusion

In this paper, we show that by training transformers over generated datasets of mathematical problems, advanced and complex computations can be learned, and qualitative and numerical tasks performed with high accuracy. Our models have no built-in mathematical knowledge, and learn from examples only. However, solving problems with high accuracy does not mean that our models have learned the techniques we use to compute their solutions. Problems such as non-autonomous control involve long and complex chains of computations, which some of the smaller models we used could probably not handle.

Most probably, our models learn shortcuts that allow them to solve specific problems, without having to learn or understand their theoretical background. Such a situation is common in everyday life. Most of us learn and use language without understanding its rules. On many practical subjects, we have tacit knowledge and know more than we can tell (Polanyi and Sen (2009)). This may be the way neural networks learn advanced mathematics. Understanding what these shortcuts are, how neural networks discover them, and how they can impact mathematical practice, is a subject for future research.

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

# A    EXAMPLES OF COMPUTATIONS

## A.1    STEP BY STEP EXAMPLE : AUTONOMOUS CONTROL

To measure whether the system

$$\frac{dx_1(t)}{dt} = \sin(x_1^2) + \log(1 + x_2) + \frac{\mathrm{atan}(ux_1)}{1 + x_2}$$

$$\frac{dx_2(t)}{dt} = x_2 - e^{x_1 x_2},$$

is controllable at a point $x_e$, with asymptotic control $u_e$, using Kalman condition we need to

1. differentiate the system with respect to its internal variables, obtain the Jacobian $A(x, u)$

$$A(x, u) = \begin{pmatrix} 2x_1 \cos(x_1^2) + \frac{u(1+x_2)^{-1}}{1+u^2 x_1^2} & (1 + x_2)^{-1} - \frac{\mathrm{atan}(ux_1)}{(1+x_2)^2} \\ -x_2 e^{x_1 x_2} & 1 - x_1 e^{x_1 x_2} \end{pmatrix}$$

2. differentiate the system with respect to its control variables, obtain a matrix $B(x, u)$

$$B(x, u) = \begin{pmatrix} x_1 ((1 + u^2 x_1^2)(1 + x_2))^{-1} \\ 0 \end{pmatrix}$$

3. evaluate $A$ and $B$ in $x_e = [0.5]$, $u_e = 1$

$$A(x_e, u_e) = \begin{pmatrix} 1.50 & 0.46 \\ -0.64 & 0.36 \end{pmatrix}, \quad B(x_e, u_e) = \begin{pmatrix} 0.27 \\ 0 \end{pmatrix}$$

4. calculate the controllability matrix given by (2).

$$C = [B, AB]((x_e, u_e)) = \left[ \begin{pmatrix} 0.27 \\ 0 \end{pmatrix}, \begin{pmatrix} 1.50 & 0.46 \\ -0.64 & 0.36 \end{pmatrix} \begin{pmatrix} 0.27 \\ 0 \end{pmatrix} \right] = \begin{pmatrix} 0.27 & 0.40 \\ 0 & -0.17 \end{pmatrix}$$

5. output $n - d$, with $d$ the rank of the controllability matrix, the system is controllable if $n - d = 0$

$$n - \mathrm{rank}(C) = 2 - 2 = 0 : \text{ System is controllable in } (x_e = [0.5], u_e = 1)$$

6. (optionally) if $n - d = 0$, compute the control feedback matrix $K$ as in (3)

$$K = \begin{pmatrix} -22.8 & 44.0 \end{pmatrix}.$$

## A.2    STEP BY STEP EXAMPLE: STABILITY OF LINEAR PDE

To find the existence and behavior at infinite time of a solution, given a differential operator $D_x$ and an initial condition $u_0$ we proceed as follows

1. find the Fourier polynomial $f(\xi)$ associated to $D_x$

$$D_x = 2\partial_{x_0}^2 + 0.5\partial_{x_1}^2 + \partial_{x_2}^4 - 7\partial_{x_0, x_1}^2 - 1.5\partial_{x_1}\partial_{x_2}^2,$$
$$f(\xi) = -4\pi\xi_0^2 - \pi\xi_1^2 + 2\pi\xi_2^4 + 14\pi\xi_0\xi_1 + 3i\pi\xi_1\xi_2^2$$

2. find the Fourier transform $\tilde{u}_0(\xi)$ of $u_0$

$$u_0(x) = e^{-3ix_2} x_0^{-1} \sin(x_0) e^{2.5ix_1} e^{-x_2^2},$$
$$\widetilde{u}_0(\xi) = \pi^{3/2} \mathbf{1}_{[-(2\pi)^{-1}, (2\pi)^{-1}]}(\xi_0)\delta_0(\xi_1 - 2.5(2\pi)^{-1})e^{-\pi^2(\xi_2 + 3(2\pi)^{-1})^2}$$

3. find the set $\mathcal{F}$ of frequency $\xi$ for which $\tilde{u}_0(\xi) \neq 0$

$$\mathcal{F} = [-(2\pi)^{-1}, (2\pi)^{-1}] \times \{2.5(2\pi)^{-1}\} \times (-\infty, +\infty)$$

4. minimize $f(\xi)$ on $\mathcal{F}$

$$\min_{\mathcal{F}}(f(\xi)) = -22.6$$

5. output $(0,0)$ if this minimum is infinite, $(1,0)$ is finite and negative, $(1,1)$ if finite and positive. (optionally) output $\mathcal{F}$

Out $= (1,0)$ : there exists a solution $u$ ; it does not vanish at $t \to +\infty$

## A.3  EXAMPLES OF INPUTS AND OUTPUTS

### A.3.1  LOCAL STABILITY

| System | Speed of convergence at $x_e = [0.01]$ |
|---|---|
| $\begin{cases} \frac{d}{dt}x_0 = -\frac{x_1}{\text{atan}\,(8x_0x_2)} + \frac{0.01}{\text{atan}\,(0.0008)} \\[2mm] \frac{d}{dt}x_1 = -\cos{(9x_0)} + \cos{(0.09)} \\[2mm] \frac{d}{dt}x_2 = x_0 - \sqrt{x_1 + x_2} - 0.01 + 0.1\sqrt{2} \end{cases}$ | $-1250$ |
| $\begin{cases} \frac{d}{dt}x_0 = -\frac{2x_2}{x_0 - 2x_2(x_1-5)} + 0.182 \\[2mm] \frac{d}{dt}x_1 = (x_1 + (x_2 - e^{x_1})(\tan{(x_0)} + 3))(\log{(3)} + i\pi) \\ \qquad\quad +3.0\log{(3)} + 3.0i\pi \\[2mm] \frac{d}{dt}x_2 = \text{asin}\left(x_0 \log\left(-\frac{4}{x_1}\right)\right) - \text{asin}\,(0.06 + 0.01i\pi) \end{cases}$ | $-0.445$ |
| $\begin{cases} \frac{d}{dt}x_0 = e^{x_1 + e^{-\sin\left(x_0 - e^2\right)}} - 1.01e^{e^{-\sin\left(0.01 - e^2\right)}} \\[2mm] \frac{d}{dt}x_1 = 0.06 - 6x_1 \\[2mm] \frac{d}{dt}x_2 = -201 + \frac{x_0 + 2}{x_0^2 x_2} \end{cases}$ | $6.0$ (locally stable) |
| $\begin{cases} \frac{d}{dt}x_0 = x_2 e^{-x_1}\sin{(x_1)} - 9.9 \cdot 10^{-5} \\[2mm] \frac{d}{dt}x_1 = 7.75.10^{-4} - \frac{e^{x_2}\,\text{atan}\,(\text{atan}\,(x_1))}{4e^{x_2} + 9} \\[2mm] \frac{d}{dt}x_2 = (x_1 - \text{asin}\,(9))\,e^{-\frac{x_0}{\log{(3)} + i\pi}} \\ \qquad\quad - (0.01 - \text{asin}\,(9))\,e^{-\frac{0.01}{\log{(3)} + i\pi}} \end{cases}$ | $-0.0384$ |
| $\begin{cases} \frac{d}{dt}x_0 = -\frac{x_0\left(7 - \sqrt[4]{7}\sqrt{i}\right)}{9} - x_1 + 0.0178 - 0.00111\sqrt[4]{7}\sqrt{i} \\[2mm] \frac{d}{dt}x_1 = -0.000379 + e^{-\frac{63}{\cos{((x_2-9)\,\text{atan}\,(x_1))} + 7}} \\[2mm] \frac{d}{dt}x_2 = -x_0 - x_1 + \text{asin}\left(\cos{(x_0)} + \frac{x_2}{x_0}\right) \\ \qquad\quad -1.55 + 1.32i \end{cases}$ | $3.52.10^{-11}$ (locally stable) |

### A.3.2 Controllability: autonomous systems

| Autonomous system | Dimension of uncontrollable space at $x_e = [0.5]$, $u_e = [0.5]$ |
|---|---|
| $\begin{cases} \frac{dx_0}{dt} = -\operatorname{asin}\left(\frac{x_1}{9} - \frac{4\tan(\cos(10))}{9}\right) \\ \qquad - \operatorname{asin}\left(\frac{4\tan(\cos(10))}{9} - 0.0556\right) \\[2mm] \frac{dx_1}{dt} = u - x_2 + \log\left(10 + \frac{\tan(x_1)}{u+x_0}\right) - 2.36 \\[2mm] \frac{dx_2}{dt} = 2x_1 + x_2 - 1.5 \end{cases}$ | 0 (controllable) |
| $\begin{cases} \frac{dx_0}{dt} = u - \operatorname{asin}(x_0) - 0.5 + \frac{\pi}{6} \\[2mm] \frac{dx_1}{dt} = x_0 - x_1 + 2x_2 + \operatorname{atan}(x_0) - 1.46 \\[2mm] \frac{dx_2}{dt} = \frac{5x_2}{\cos(x_2)} - 2.85 \end{cases}$ | 1 |
| $\begin{cases} \frac{dx_0}{dt} = 6u + 6x_0 - \frac{6x_1}{x_0} \\[2mm] \frac{dx_1}{dt} = 0.75 + x_1^2 - \cos(u - x_2) \\[2mm] \frac{dx_2}{dt} = -x_0^2 + x_0 + \log(e^{x_2}) - 0.75 \end{cases}$ | 2 |
| $\begin{cases} \frac{dx_0}{dt} = +x_0\left(\cos\left(\frac{u}{x_0+2x_2}\right) + \frac{\operatorname{asin}(u)}{x_1}\right) \\ \qquad -0.5\cos\left(\frac{1}{3}\right) - \frac{\pi}{6} \\[2mm] \frac{dx_1}{dt} = \frac{\pi x_1}{4(x_2+4)} - \frac{\pi}{36} \\[2mm] \frac{dx_2}{dt} = 2.5 - 108e^{0.5} - 12x_0x_2 + x_1 + 108e^u \end{cases}$ | 0 (controllable) |
| $\begin{cases} \frac{dx_0}{dt} = -10\sin\left(\frac{3x_0}{\log(8)} - 22\right) - 6.54 \\[2mm] \frac{dx_1}{dt} = \sin\left(9 + \frac{-x_1-4}{8x_2}\right) - 1 \\[2mm] \frac{dx_2}{dt} = 4\tan\left(\frac{4x_0}{u}\right) - 4\tan(4) \end{cases}$ | 1 |

### A.3.3 CONTROLLABILITY: NON-AUTONOMOUS SYSTEMS

| Non-autonomous system | Local controllability at $x_e = [0.5]$, $u_e = [0.5]$ |
|---|---|
| $\begin{cases} \frac{dx_0}{dt} = (x_2 - 0.5)\,e^{-\operatorname{asin}(8)} \\ \frac{dx_1}{dt} = e^{t+0.5} - e^{t+x_1} + \frac{-x_1 + e^{\frac{x_0}{u}}}{x_2} + 1 - 2e \\ \frac{dx_2}{dt} = t(x_2 - 0.5)\left(\operatorname{asin}(6) + \sqrt{\tan(8)}\right) \end{cases}$ | False |
| $\begin{cases} \frac{dx_0}{dt} = \frac{\operatorname{atan}(\sqrt{x_2})}{x_0 - 1} - 2\operatorname{atan}\left(\frac{\sqrt{2}}{2}\right) \\ \frac{dx_1}{dt} = -\frac{u}{-\sqrt{x_0}x_1 + 3} + x_2 + \log(x_0) \\ \qquad + \log(2) - 0.5 + (1/(6 - \sqrt{2})) \\ \frac{dx_2}{dt} = -70t(x_0 - 0.5) \end{cases}$ | False |
| $\begin{cases} \frac{dx_0}{dt} = \frac{x_0 + 7}{\sin(x_0 e^u) + 3} \\ \frac{dx_1}{dt} = -\frac{9x_2 e^{-\sin\left(\sqrt{\log(x_1)}\right)}}{x_0} \\ \frac{dx_2}{dt} = t + \operatorname{asin}(tx_2 + 4) \end{cases}$ | False |
| $\begin{cases} \frac{dx_0}{dt} = 0.5 - x_2 + \tan(x_0) - \tan(0.5) \\ \frac{dx_1}{dt} = \frac{t}{x_1(t + \cos(x_1(t+u)))} - \frac{t}{0.5(t + \cos(0.5t + 0.25))} \\ \frac{dx_2}{dt} = 2.75 - x_0(u + 4) - x_0 \end{cases}$ | True |
| $\begin{cases} \frac{dx_0}{dt} = u(u - x_0 - \tan(8)) + 0.5(\tan(8)) \\ \frac{dx_1}{dt} = -\frac{6t\left(-2 + \frac{\pi}{2}\right)}{x_0 x_1} - 12t(4 - \pi) \\ \frac{dx_2}{dt} = -7(u - 0.5) - 7\tan(\log(x_2)) \\ \qquad + 7\tan(\log(0.5)) \end{cases}$ | True |

### A.3.4 STABILITY OF PARTIAL DIFFERENTIAL EQUATIONS USING FOURIER TRANSFORM

| PDE $\partial_t u + D_x u = 0$ and initial condition | Existence of a solution, $u \to 0$ at $t \to +\infty$ |
|---|---|
| $\begin{cases} D_x = 2\partial_{x_0}\left(2\partial_{x_0}^4\partial_{x_2}^4 + 3\partial_{x_1}^3 + 3\partial_{x_1}^2\right) \\ \\ u_0 = \delta_0(-18x_0)\delta_0(-62x_2)e^{89ix_0 - 8649x_1^2 + 89ix_1 - 59ix_2} \end{cases}$ | False , False |
| $\begin{cases} D_x = -4\partial_{x_0}^4 - 5\partial_{x_0}^3 - 6\partial_{x_0}^2\partial_{x_1}^2\partial_{x_2}^2 + 3\partial_{x_0}^2\partial_{x_1} - 4\partial_{x_1}^6 \\ \\ u_0 = (162x_0 x_2)^{-1}\left(e^{i(-25x_0 + 96x_2)}\sin(54x_0)\sin(3x_2)\right) \end{cases}$ | True , False |
| $\begin{cases} D_x = \partial_{x_1}\left(4\partial_{x_0}^5\partial_{x_1} + 4\partial_{x_0}^2 - 9\partial_{x_0}\partial_{x_2}^6 \right. \\ \left. + 2\partial_{x_1}^3\partial_{x_2}^5 - 4\partial_{x_1}^3\partial_{x_2}^4 - 2\partial_{x_2}\right) \\ \\ u_0 = (33x_0)^{-1}\left(e^{86ix_0 - 56ix_1 - 16x_2^2 + 87ix_2}\sin(33x_0)\right) \end{cases}$ | True , False |
| $\begin{cases} D_x = -6\partial_{x_0}^7\partial_{x_2}^2 + \partial_{x_0}^5\partial_{x_2}^6 - 9\partial_{x_0}^4\partial_{x_1}^2 - 9\partial_{x_0}^4\partial_{x_2}^4 \\ + 7\partial_{x_0}^2\partial_{x_2}^6 + 4\partial_{x_0}^2\partial_{x_2}^5 - 6\partial_{x_1}^6 \\ \\ u_0 = \delta_0(88x_1)e^{-2x_0(2312x_0 + 15i)} \end{cases}$ | True , True |

# B   MATHEMATICAL DEFINITIONS AND THEOREMS

## B.1   NOTIONS OF STABILITY

Let us consider a system

$$\frac{dx(t)}{dt} = f(x(t)). \tag{7}$$

$x_e$ is an attractor, if there exists $\rho > 0$ such that

$$|x(0) - x_e| < \rho \Longrightarrow \lim_{t \to +\infty} x(t) = x_e. \tag{8}$$

But, counter intuitive as it may seem, this is not enough for asymptotic stability to take place.

**Definition B.1.** *We say that $x_e$ is a locally (asymptotically) stable equilibrium if the two following conditions are satisfied:*

*(i) $x_e$ is a stable point, i.e. for every $\varepsilon > 0$, there exists $\eta > 0$ such that*

$$|x(0) - x_e| < \eta \Longrightarrow |x(t) - x_e| < \varepsilon, \ \forall \, t \geq 0. \tag{9}$$

*(ii) $x_e$ is an attractor, i.e. there exists $\rho > 0$ such that*

$$|x(0) - x_e| < \rho \Longrightarrow \lim_{t \to +\infty} x(t) = x_e. \tag{10}$$

In fact, the SMT of Subsection 3.1 deals with an even stronger notion of stability, namely the exponential stability defined as follows:

**Definition B.2.** *We say that $x_e$ is an exponentially stable equilibrium if $x_e$ is locally stable equilibrium and, in addition, there exist $\rho > 0$, $\lambda > 0$, and $M > 0$ such that*

$$|x(0) - x_e| < \rho \Longrightarrow |x(t)| \leq Me^{-\lambda t}|x(0)|.$$

In this definition, $\lambda$ is called the exponential convergence rate, which is the quantity predicted in our first task. Of course, if $x_e$ is locally exponentially stable it is in addition locally asymptotically stable.

## B.2   CONTROLLABILITY

We give here a proper mathematical definition of controllability. Let us consider a non-autonomous system

$$\frac{dx(t)}{dt} = f(x(t), u(t), t), \tag{11}$$

such that $f(x_e, u_e) = 0$.

**Definition B.3.** *Let $\tau > 0$, we say that the nonlinear system (11) is locally controllable at the equilibrium $x_e$ in time $\tau$ with asymptotic control $u_e$ if, for every $\varepsilon > 0$, there exists $\eta > 0$ such that, for every $(x_0, x_1) \in \mathbb{R}^n \times \mathbb{R}^n$ with $|x_0 - x_e| \leq \eta$ and $|x_1 - x_e| \leq \eta$ there exists a trajectory $(x, u)$ such that*

$$\begin{aligned} x(0) = x_0, \quad x(\tau) = x_1 \\ |u(t) - u_e| \leq \varepsilon, \quad \forall \, t \in [0, \tau]. \end{aligned} \tag{12}$$

An interesting remark is that if the system is autonomous, the local controllability does not depend on the time $\tau$ considered, which explains that it is not precised in Theorem 3.2.

### B.3 Tempered distribution

We start by recalling the multi-index notation: let $\alpha = (\alpha_1, ..., \alpha_n) \in \mathbb{N}^n$, $x \in \mathbb{R}^n$, and $f \in C^\infty(\mathbb{R}^n)$, we denote

$$
\begin{aligned}
x^\alpha &= x_1^{\alpha_1} \times \cdots \times x_n^{\alpha_n} \\
\partial_x^\alpha f &= \partial_{x_1}^{\alpha_1} \ldots \partial_{x_n}^{\alpha_n} f.
\end{aligned}
\tag{13}
$$

$\alpha$ is said to be a multi-index and $|\alpha| = \sum_{i=1}^{n} |\alpha_i|$. Then we give the definition of the Schwartz functions:

**Definition B.4.** *A function $\phi \in C^\infty$ belongs to the Schwartz space $\mathcal{S}(\mathbb{R}^n)$ if, for any multi-index $\alpha$ and $\beta$,*

$$
\sup_{x \in \mathbb{R}^n} |x^\alpha \partial_x^\beta \phi| < +\infty.
\tag{14}
$$

Finally, we define the space of tempered distributions:

**Definition B.5.** *A tempered distribution $\phi \in \mathcal{S}'(\mathbb{R}^n)$ is a linear form $u$ on $\mathcal{S}(\mathbb{R}^n)$ such that there exists $p > 0$ and $C > 0$ such that*

$$
|\langle u, \phi \rangle| \leq C \sum_{|\alpha|, |\beta| < p} \sup_{x \in \mathbb{R}^n} |x^\alpha \partial_x^\beta \phi|, \quad \forall \, \phi \in \mathcal{S}(\mathbb{R}^n).
\tag{15}
$$

### B.4 Proofs of theorems

#### B.4.1 Analysis of Problem 2

The proofs of Theorem 3.2, of validity of the feedback matrix given by the expression (3), and of the extension of Theorem 3.2 to the non-autonomous system given by condition (4) can be found in Coron (2007). We give here the key steps of the proof for showing that the matrix $K$ given by (3) is a valid feedback matrix to illustrate the underlying mechanisms:

- Setting $V(x(t)) = x(t)^{tr} C_T^{-1} x(t)$, where $x$ is solution to $x'(t) = f(x, u_e + K.(x - x_e))$, and

$$
C_T = \left( e^{-AT} \left[ \int_0^T e^{-At} BB^{tr} e^{-A^{tr} t} dt \right] e^{-A^{tr} T} \right).
\tag{16}
$$

- Showing, using the form of $C_T$, that

$$
\frac{d}{dt}(V(x(t))) = -|B^{tr} C_T^{-1} x(t)|^2 - |B^{tr} e^{-TA^{tr}} C_T^{-1} x(t)|^2
$$

- Showing that, if for any $t \in [0, T]$, $|B^{tr} C_T^{-1} x(t)|^2 = 0$, then for any $i \in \{0, ..., n-1\}$,

$$
x^{tr} C_T^{-1} A^i B = 0, \quad \forall \, t \in [0, T].
$$

- Deducing from the controllability condition (2), that

$$
x(t)^{tr} C_T^{-1} = 0, \quad \forall \, t \in [0, T].
$$

and therefore from the invertibility of $C_T^{-1}$,

$$
x(t) = 0, \quad \forall \, t \in [0, T].
$$

- Concluding from the previous and LaSalle invariance principle that the system is locally exponentially stable.

#### B.4.2 Analysis of Problem 3

In this section we prove Proposition 3.1. We study the problem

$$
\partial_t u + \sum_{|\alpha| \leq k} a_\alpha \partial_x^\alpha u = 0 \text{ on } \mathbb{R}_+ \times \mathbb{R}^n,
\tag{17}
$$

with initial condition

$$u(0, \cdot) = u_0 \in \mathcal{S}'(\mathbb{R}^n), \tag{18}$$

and we want to find a solution $u \in C^0([0, T], \mathcal{S}'(\mathbb{R}^n))$.

Denoting $\widetilde{u}$ the Fourier transform of $u$ with respect to $x$, the problem is equivalent to

$$\partial_t \widetilde{u}(t, \xi) + \sum_{|\alpha| \leq k} a_\alpha (i\xi)^\alpha \widetilde{u}(t, \xi) = 0, \tag{19}$$

with initial condition $\widetilde{u}_0 \in \mathcal{S}(\mathbb{R}^n)$. As the only derivative now is with respect to time, we can check that

$$\widetilde{u}(t, \xi) = \widetilde{u}_0(\xi) e^{-f(\xi)t}, \tag{20}$$

where $f(\xi) = \sum_{|\alpha| \leq k} a_\alpha (i\xi)^\alpha$, is a weak solution to (19) belonging to the space $C^0([0, +\infty), \mathcal{D}'(\mathbb{R}^n))$. Indeed, first of all we can check that for any $t \in [0, +\infty)$, $\xi \to \exp(-f(\xi)t)$ is a continuous function and $\widetilde{u}_0$ belongs to $\mathcal{S}'(\mathbb{R}^n) \subset \mathcal{D}'(\mathbb{R}^n)$, thus $\widetilde{u}(t, \cdot)$ belongs to $\mathcal{D}'(\mathbb{R}^n)$. Besides, $t \to e^{-f(\xi)t}$ is a $C^\infty$ function whose derivative in time are of the form $P(\xi) e^{-f(\xi)t}$ where $P(\xi)$ is a polynomial function. $\widetilde{u}$ is continuous in time and $\widetilde{u} \in C^0([0, +\infty), \mathcal{D}'(\mathbb{R}^n))$. Now we check that it is a weak solution to (19) with initial condition $\widetilde{u}_0$. Let $\phi \in C_c^\infty([0, +\infty) \times \mathbb{R}^n)$ the space of smooth functions with compact support, we have

$$\begin{aligned}
& - \langle \widetilde{u}, \partial_t \phi \rangle + \sum_{|\alpha| \leq k} a_\alpha (i\xi)^\alpha \langle \widetilde{u}, \phi \rangle + \langle \widetilde{u}_0, \phi \rangle \\
= & - \langle \widetilde{u}_0, \partial_t (e^{-\overline{f(\xi)}t} \phi) \rangle - \langle \widetilde{u}_0, \overline{f(\xi)} e^{-\overline{f(\xi)}t} \phi \rangle + \langle \widetilde{u}_0, e^{-\overline{f(\xi)}t} \overline{f(\xi)} \phi \rangle + \langle \widetilde{u}_0, \phi \rangle \\
= & \, 0.
\end{aligned} \tag{21}$$

Hence, $u$ defined by (20) is indeed a weak solution of (19) in $C^0([0, +\infty), \mathcal{D}'(\mathbb{R}^n))$. Now, this does not answer our question as this only tells us that at time $t > 0$, $u(t, \cdot) \in \mathcal{D}'(\mathbb{R}^n)$ which is a less regular space than the space of tempered distribution $\mathcal{S}'(\mathbb{R}^n)$. In other words, at $t = 0$, $\widetilde{u} = \widetilde{u}_0$ has a higher regularity by being in $\mathcal{S}'(\mathbb{R}^n)$ and we would like to know if equation (19) preserves this regularity. This is more than a regularity issue as, if not, one cannot define a solution $u$ as the inverse Fourier Transform of $\widetilde{u}$ because such function might not exist. Assume now that there exists a constant $C$ such that

$$\forall \xi \in \mathbb{R}^n, \ \widetilde{u}_0(\xi) = 0 \ \text{ or } \ \mathrm{Re}(f(\xi)) > C. \tag{22}$$

$$\forall \, \xi \in \mathbb{R}^n, \ \mathbf{1}_{\mathrm{supp}(\widetilde{u}_0)} e^{-f(\xi)t} \leq e^{-Ct}. \tag{23}$$

This implies that, for any $t > 0$, $\widetilde{u} \in \mathcal{S}'(\mathbb{R}^n)$. Besides, defining for any $p \in \mathbb{N}$,

$$\mathcal{N}_p(\phi) = \sum_{|\alpha|, |\beta| < p} \sup_{\xi \in \mathbb{R}^n} |\xi^\alpha \partial_\xi^\beta \phi(\xi)|, \tag{24}$$

then for $t_1, t_2 \in [0, T]$,

$$\mathcal{N}_p((e^{-f(\xi)t_1} - e^{-f(\xi)t_2})\phi) = \sum_{|\alpha|, |\beta| < p} \sup_{\xi \in \mathbb{R}^n} |\xi^\alpha P_\beta(\xi, \phi)|, \tag{25}$$

where $P_\beta(\xi, \phi)$ is polynomial with $f(\xi)$, $\phi(\xi)$, and their derivatives of order strictly smaller than $p$. Besides, each term of this polynomial tend to 0 when $t_1$ tends to $t_2$ on $\mathrm{supp}(\widetilde{u}_0)$, the set of frequency of $u_0$. Indeed, let $\beta_1$ be a multi-index, $k \in \mathbb{N}$, and $Q_i(\xi)$ be polynomials in $\xi$, where $i \in \{0, ..., k\}$.

$$\begin{aligned}
& \left| \mathbf{1}_{\mathrm{supp}(u_0)} \partial_\xi^{\beta_1} \phi(\xi) \left( \sum_{i=0}^{k} Q_i(\xi) t_1^i e^{-f(\xi)t_1} - Q_i(\xi) t_2^i e^{-f(\xi)t_2} \right) \right| \\
& \leq \sum_{i=0}^{k} \max_{\mathrm{supp}(\widetilde{u}_0)} \left| t_1^i e^{-f(\xi)t_1} - t_2^i e^{-f(\xi)t_2} \right| \max_{\xi \in \mathbb{R}^n} \left| \partial_\xi^{\beta_1} \phi(\xi) Q_i(\xi, t) \right|.
\end{aligned} \tag{26}$$

From (22), the time-dependant terms in the right-hand sides converge to 0 when $t_1$ tends to $t_2$. This implies that $u \in C^0([0,T], \mathcal{S}'(\mathbb{R}^n))$. Finally let us show the property of the behavior at infinity. Assume that $C > 0$, one has, for any $\phi \in S(\mathbb{R}^n)$

$$\langle \widetilde{u}(t,\cdot), \phi \rangle = \langle \widetilde{u}_0, \mathbf{1}_{\text{supp}(\widetilde{u}_0)} e^{-\overline{f(\xi)}t} \phi \rangle. \tag{27}$$

Let us set $g(\xi) = e^{-\overline{f(\xi)}t} \phi(\xi)$, one has for two multi-index $\alpha$ and $\beta$

$$|\xi^\alpha \partial_\xi^\beta g(\xi)| \leq |\xi^\alpha Q(\xi) e^{-f(\xi)t}|, \tag{28}$$

where $Q$ is a sum of polynomials, each multiplied by $\phi(\xi)$ or one of its derivatives. Thus $\xi^\alpha Q(\xi)$ belongs to $\mathcal{S}(\mathbb{R}^n)$ and therefore, from assumption (22),

$$|\xi^\alpha \partial_\xi^\beta g(\xi)| \mathbf{1}_{\text{supp}(u_0)} \leq \max_{\xi \in \mathbb{R}^n} |\xi^\alpha Q(\xi)| e^{-Ct}, \tag{29}$$

which goes to 0 when $t \to +\infty$. This imply that $\widetilde{u}(t,\cdot) \to 0$ in $\mathcal{S}'(\mathbb{R}^n)$ when $t \to +\infty$, and hence $u(t,\cdot) \to 0$. This ends the proof of Proposition 3.1.

Let us note that one could try to find solutions with lower regularity, where $u$ is a distribution of $\mathcal{D}'(\mathbb{R}_+ \times \mathbb{R}^n)$, and satisfies the equation

$$\partial_t u + \sum_{|\alpha| \leq k} a_\alpha \partial_x^\alpha u = \delta_{t=0} u_0 \text{ on } \mathbb{R}_+ \times \mathbb{R}^n. \tag{30}$$

This could be done using for instance Malgrange-Erhenpreis theorem, however, studying the behavior at $t \to +\infty$ may be harder mathematically, hence this approach was not considered in this paper.

## C   ADDITIONAL EXPERIMENTS

### C.1   PREDICTION OF SPEED OF CONVERGENCE WITH HIGHER PRECISION

In Section 5.1, $\lambda$ is predicted with a 10% margin error. Prediction of $\lambda$ to better accuracy can be achieved by training models on data rounded to 2, 3 or 4 significant digits, and measuring the number of exact predictions on the test sample. Overall, we predict $\lambda$ with two significant digits in 59.2% of test cases. Table 8 summarizes the results for different precisions (for transformers with 6 layers and a dimensionality of 512).

Table 8: Exact prediction of local convergence speed to given precision.

|          | Degree 2 | Degree 3 | Degree 4 | Degree 5 | Degree 6 | Overall |
|----------|----------|----------|----------|----------|----------|---------|
| 2 digits | 83.5     | 68.6     | 55.6     | 48.3     | 40.0     | 59.2    |
| 3 digits | 75.3     | 53.2     | 39.4     | 33.4     | 26.8     | 45.7    |
| 4 digits | 62.0     | 35.9     | 25.0     | 19.0     | 14.0     | 31.3    |

### C.2   OUT-OF-DISTRIBUTION GENERALIZATION

In all our experiments, trained models are tested on held-out samples generated with the same procedure as the training data, and our results prove that the model can generalize out of the training data. However, training and test data come from the same statistical distribution (iid). This would not happen in practical cases: problems would come from some unknown distribution over problem space. Therefore, it is interesting to investigate how the model performs when the test set follows a different statistical distribution. This provides insight about how learned properties generalize, and may indicate specific cases over which the model struggles.

To this purpose, we modified the data generator to produce new test datasets for end to end stability prediction (section 5.1). Four modifications were considered:

1. **Unary operators:** varying the distribution of operators in the system. In the training data, unary operators are selected at random from a set of nine, three trigonometric functions, three inverse trigonometric functions, logarithm and exponential, and square root (the four basic operations are always present). In this set of experiments, we generated four test sets, without trigonometric functions, without logs and exponentials, only with square roots, and with a different balance of operators (mostly square roots).

2. **Variables and integers:** varying the distribution of variables in the system. In the training data, 30% of the leaves are numbers, the rest variables. We changed this probability to 0.1, 0.5 and 0.7. This has no impact on expression length, but higher probabilities make the Jacobians more sparse.

3. **Expression lengths:** making expressions longer than in the train set. In the training data, for a system of $n$ equations, we generate functions with 3 to $2n + 3$ operators. In this experiments, we tried functions between $n + 3$ and $3n + 3$ and $2n + 3$ and $4n + 3$. This means that the test sequences are, on average, much longer that those seen at training, a known weakness of sequence to sequence models.

4. **Larger degree:** our models were trained on systems with 2 to 5 equations, we tried to test it on systems with 6 equations. Again, this usually proves difficult for transformers.

Note that the two first sets of experiments feature out-of-distribution tests, exploring different distributions over the same problem space as the training data. The two last sets, on the other hand, explore a different problem space, featuring longer sequences.

Table 9 presents the results of these experiments. Changing the distribution of operators, variables and integers has little impact on accuracy, up to two limiting cases. First, over systems of degree five (the largest in our set, and more difficult for the transformers) change in operator distribution has a small adverse impact on performance (but not change in variable distribution). Second, which the proportion of integers become very large, and therefore Jacobians become very sparse, the degree of the systems has less impact on performance. But overall results remain over 95%, and the model proves to be very resistant to changes in distribution over the same problem space.

Over systems with longer expressions, overall accuracy tends to decreases. Yet, systems of two or three equations are not affected by a doubling of the number of operators (and sequence length), compared to the training data. Most of the loss in performance concentrates on larger degrees, which suggests that it results from the fact that the transformer is presented at test time with much longer sequences that what it saw at training. In any case, all results but one are well above the fastText baseline (60.5%).

When tested on systems with six equations, the trained model predicts stability in 78.7% of cases. This is a very interesting result, where the model is extrapolating out of the problem space (i.e. no system of six equations have been seen during training) with an accuracy well above chance level, and the fastText baseline.

Table 9: **End to end stability: generalization over different test sets.**

|  | Overall | Degree 2 | Degree 3 | Degree 4 | Degree 5 |
|---|---|---|---|---|---|
| Baseline: training distribution | 96.4 | 98.4 | 97.3 | 95.9 | 94.1 |
| Unary operators: no trigs | 95.7 | 98.8 | 97.3 | 95.5 | 91.2 |
| Unary operators: no logs | 95.3 | 98.2 | 97.1 | 95.2 | 90.8 |
| Unary operators: no logs and trigs | 95.7 | 98.8 | 97.7 | 95.2 | 91.0 |
| Unary operators: less logs and trigs | 95.9 | 98.8 | 96.8 | 95.0 | 93.1 |
| Variables and integers: 10% integers | 96.1 | 98.6 | 97.3 | 94.7 | 93.8 |
| Variables and integers: 50% integers | 95.6 | 97.8 | 96.7 | 94.3 | 93.1 |
| Variables and integers: 70% integers | 95.7 | 95.7 | 95.9 | 95.7 | 95.5 |
| Expression lengths: n+3 to 3n+3 | 89.5 | 96.5 | 92.6 | 90.0 | 77.9 |
| Expression lengths: 2n+3 to 4n+3 | 79.3 | 93.3 | 88.3 | 73.4 | 58.2 |
| System degree: degree 6 | 78.7 |  |  |  |  |

## D  MODEL AND PROBLEM SPACE

### D.1  MODEL ARCHITECTURE

The networks used in this paper are very close to the one described in Vaswani et al. (2017). They use an encoder/decoder architecture. The encoder stack contains 6 transformer layers, each with a 8 head self-attention layer, a normalization layer, and a one layer feed forward network with 2048 hidden units. Inputs is fed through trainable embedding and positional embedding, and the encoder stack learns a representation of dimension 512. The decoder contains 6 transformer layers, each with a (8-head) self-attention layer, a cross attention (pointing to the encoder output) layer, normalization and feed forward linear layer. Representation dimension is the same as the encoder (512). The final output is sent to a linear layer that decodes the results.

The training loss is the cross entropy between the model predicted output and actual result from the dataset. During training, we use the Adam optimizer, with a learning rate of 0.0001 and scheduling (as in Vaswani et al. (2017)). Mini-batch size varies from one problem to the other, typically between 32 and 128 examples.

During training, we use 8 GPU. The model is distributed across GPUs, so that all of them have access to the same shared copy of the model. At each iteration, every GPU processes an independently generated batch, and the optimizer updated the model weights using the gradients accumulated by all GPU. Overall, this is equivalent to training on a single GPU, but with 8 times larger batches.

### D.2  MODEL BEHAVIOR AND ATTENTIONS HEADS

We tried to analyze model behavior by looking at the attention heads and the tokens the models focus on when it predicts a specific sequence. As each head attends many more tokens than in usual natural language tasks, and to improve visualization, we tried to reduce the number of hidden states a head can attend by using a top-k on the attention weights, but this deteriorated the performance, and we did not investigate more in this direction. We also ran a sequence-to-sequence model without attention, so that each input equation is mapped to a fixed-sized representation. We then fed a set of input equations into the model, and used a t-SNE visualization to see whether we can observe clusters of equations. What we observed is mainly that equations with nearby representations have similar length / tokens. However, even embeddings in similar locations can lead to different decoded sequences. The relevance of the representations built in the encoder depends on how the computation is split between the encoder and the decoder. If the decoder does the majority of the work, encoder representations become less meaningful.

## D.3 Learning curves

Although all generated datasets included more than 50 million examples, most models were trained on less. Figure 1 shows how performance increases with the number of training examples, for the end to end stability problem (i.e. predicting whether systems of degree 2 to 5 are stable). There are twelve curves corresponding to as many experiments over shuffled versions of the dataset (i.e. different experiments used different parts of the dataset).

Overall, less than 10 million examples are needed to achieve close to optimal accuracy. Learning curves from different experiments are close, which proves the stability of the learning process.

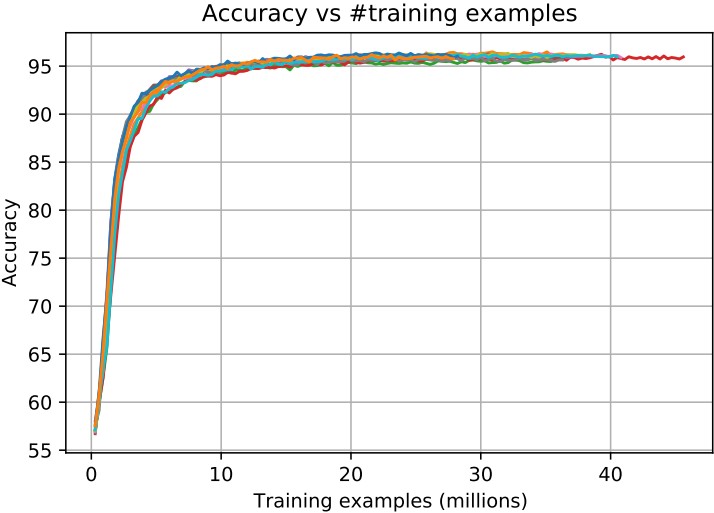

Figure 1: **End to end stability accuracy vs number of training examples.** 12 models, trained over shuffled versions of the same dataset.

## D.4 Size of the problem space

Lample and Charton (2020) provide the following formula to calculate the number of functions with $m$ operators:

$$E_0 = L$$
$$E_1 = (q_1 + q_2 L)L$$
$$(m + 1)E_m = (q_1 + 2q_2 L)(2m - 1)E_{m-1} - q_1(m - 2)E_{m-2}$$

Where $L$ is the number of possible leaves (integers or variables), and $q_1$ and $q_2$ the number of unary and binary operators. In the stability and controllability problems, we have $q_1 = 9$, $q_2 = 4$ and $L = 20 + q$, with $q$ the number of variables.

Replacing, we have, for a function with $q$ variables and $m$ operators

$$E_0(q) = 20 + q$$
$$E_1(q) = (89 + 4q)(20 + q)$$
$$(m + 1)E_m(q) = (169 + 8q)(2m - 1)E_{m-1} - 4(m - 2)E_{m-2}$$

In the stability problem, we sampled systems of $n$ functions, with $n$ variables, $n$ from 2 to 6. Functions have between 3 and $2n + 2$ operators. The number of possible systems is

$$PS_{st} = \sum_{n=2}^{6} \left( \sum_{m=3}^{2n+2} E_m(n) \right)^n > E_{14}(6)^6 \approx 3.10^{212}$$

(since $E_m(n)$ increases exponentially with $m$ and $n$, the dominant factor in the sum is the term with largest $m$ and $n$)

In the autonomous controllability problem, we generated systems with $n$ functions ($n$ between 3 and 6), and $n + p$ variables ($p$ between 1 and $n/2$). Functions had between $n + p$ and $2n + 2p + 2$ operators. The number of systems is

$$PS_{aut} = \sum_{n=3}^{6} \left( \sum_{p=1}^{n/2} \sum_{m=n+p}^{2(n+p+1)} E_m(n+p) \right)^n > E_{20}(9)^6 \approx 4.10^{310}$$

For the non-autonomous case, the number of variables in $n + p + 1$, $n$ is between 2 and 3 and $p = 1$, therefore

$$PS_{naut} = \sum_{n=2}^{3} \left( \sum_{m=n+1}^{2(n+2)} E_m(n+2) \right)^n > E_{10}(5)^3 \approx 5.10^{74}$$

Because expressions with undefinite or degenerate jacobians are skipped, the actual problem space size will be smaller by several orders of magnitude. Yet, problem space remains large enough for overfitting by memorizing problems and solutions to be impossible.

## E    Computation efficiency

### E.1    Algorithmic complexity

Let $n$ be the system degree, $p$ the number of variables and $q$ the average length (in tokens) of functions in the system. In all problems considered here, we have $p = O(n)$. Differentiating or evaluating an expression with $q$ tokens is O(q), and calculating the Jacobian of our system is $O(npq)$, i.e. $O(n^2q)$.

In the stability experiment, calculating the eigenvalues of the Jacobian will be $O(n^3)$ in most practical situations. In the autonomous controllability experiments, construction of the $n \times np$ Kalman matrix is $O(n^3p)$, and computing its rank, via singular value decomposition or any equivalent algorithm, will be $O(n^3p)$ as well. The same complexity arise for feedback matrix computations (multiplication, exponentiation and inversion are all $O(n^3)$ for a square $n$ matrix). As a result, for controllability, complexity is $O(n^4)$. Overall, the classical algorithms have a complexity of $O(n^2q)$ for Jacobian calculation, and $O(n^3)$ (stability) and $O(n^4)$ (controllability) for the problem specific computations.

Current transformer architectures are quadratic in the length of the sequence, in our case $nq$, so a transformer will be $O(n^2q^2)$ (in speed and memory usage). Therefore, the final comparison will depend on how $q$, the average length of equations, varies with $n$, the number of parameters. If $q = O(1)$ or $O(log(n))$, transformers have a large advantage over classical methods. This means sparse Jacobians, a condition often met in practice. For controllability, the advantage remains if $q = O(n^{1/2})$, and the two methods are asymptotically equivalent if $q = O(n)$.

However, current research is working on improving transformer complexity to log-linear or linear. If this happened (and there seem to be no theoretical reason preventing it), transformers would have lower asymptotic complexity in all cases.

## E.2   Computation time versus evaluation time

Table 10 compares the average time needed to solve one problem, for a trained transformer running on a GPU, and a Python implementation of the algorithms, running on a MacBook Pro.

Table 10: **Speed comparison between trained transformers and mathematical libraries.** Average time to solve one system, in seconds.

| Task | Mathematical libraries | Trained transformers |
|---|---|---|
| Stability end to end | 0.02 | 0.0008 |
| Stability largest eigenvalue | 0.02 | 0.002 |
| Controllability (autonomous) | 0.05 | 0.001 |
| Predicting a feedback matrix | 0.4 | 0.002 |

