# OpenReview forum: "Learning advanced mathematical computations from examples"
_ICLR.cc/2021/Conference — ICLR 2021 Poster_

### Official Review · AnonReviewer1 · 2020-10-24
**Review of Transformers for differential systems**

**Rating:** 6
**Confidence:** 3

**Review:**

This paper investigates the use of deep learning models, and specifically transformers, to learn mathematical properties of differential systems. Authors tackle three problems:

* Local stability: where the goal is predicting the stability of a given system at a given point, similar to Spectral Mapping Theorem.
* Control theory: where the goal is predicting controllability and computing the control feedback matrix as in Kalman condition.
* Stability of PDEs using Fourier transform: where the goal is to predict the existence and stability of a PDE given its differential operator.

Overall, this paper is well written and tackles an intersting problem with potential useful insights. I believe it can be improved by making the discussion more rigorous:

1. How general are the class of functions considered in each problem? For example, you generate random functions by sampling unary- binary trees. Does this give the guarantee that the majority of solutions for each of three problems can learned by deep learning methods?

2. How this work compares to previous ones such as (Lample and Charton 2020) that study learning symbolic mathematics using deep learning? Any distinct insight when investigating numerical and differential systems?

3. It is useful to see if generalizability of learned mathematical computation can be assessed by experiments, where training datasets are generated using some class of functions, and they are tested on a separated class of functions. Are such experiments feasible using unary- binary trees?

---

> ### Author Response · Authors · 2020-11-17
> **Response to Reviewer 1**
>
> Thank you very much for your review, here are preliminary replies to your observations.
>
> ==== How general are the class of functions considered in each problem? For example, you generate random functions by sampling unary- binary trees. Does this give the guarantee that the majority of solutions for each of three problems can learned by deep learning methods? =====
>
> This is clearly an important question, thank you for raising it. Any function that can be written using standard mathematical notation can be represented as a unary-binary tree. Therefore this process does give us the guarantee that not only the majority but even any solution of the three problems can be generated with our setting. Of course, this is only true up to the choice of the constants and the elementary operators. We chose here to include the most used elementary functions: square root, power, exp, log, sin, cos, tan, asin, acos, atan, etc. This covers most of the physical and mathematical systems studied in practice. It would still be possible to add other functions to the mix, like special functions (Bessel, Fresnel), but we chose to limit ourselves to the most commonly used operators in this paper.
>
> ===== How this work compares to previous ones such as (Lample and Charton 2020) that study learning symbolic mathematics using deep learning? Any distinct insight when investigating numerical and differential systems? =====
>
> Our transformer architectures are very similar to those used by Lample and Charton (LC henceforth). The only significant difference on this aspect is that we use scheduling of the learning rate, and do not perform beam search at inference. We also use their techniques for random function generation. That being said, our problems are in fact quite different and we believe we generalize their findings on three counts.
>
> First, we focus on a different class of problems. LC are performing machine translation: their inputs and outputs are of the same nature (sequences representing functions). The problems solved here are classification and regression, inputs are sequences representing functions, but outputs are arrays of real numbers (represented as sequences) or binary values. This complicates the work of the transformer, as it needs to learn both input and output syntax, and cannot leverage the input syntax to infer solutions.
>
> Second, LC focus on symbolic problems that have a strong connection to pattern matching. When integrating a function, one looks for specific parts in the input sequence that will be replaced by specific translations in the output (for instance, a cosine will always be integrated into a sine). Pattern matching is common in natural language processing and transformers tend to be good at it. Therefore, one might wonder whether LC results were partly due to the transformer exploiting the "linguistic" character of the symbolic maths problems they consider. The problems studied in our paper feature some symbolic calculations, but also computations which involve no pattern matching (eigenvalues, point estimations, minimization). That transformers can successfully learn such problems show that their potential for mathematical tasks may be larger than what was thought before and could be used in many areas where symbolic and numerical operations are mixed: physics, computational-biology, theoretical chemistry, etc.
>
> Finally, most LC datasets were built by generating problems from their solutions. This raises a question about how representative of the problems space their datasets are. All problems considered here were generated directly, at random.
>
> ===== It is useful to see if generalizability of learned mathematical computation can be assessed by experiments, where training datasets are generated using some class of functions, and they are tested on a separated class of functions. Are such experiments feasible using unary- binary trees? =====
>
> Thank you for this suggestion! Indeed looking at tests on different class of functions would be very interesting to test the generalizability of the learned mathematical computations. This can be done by varying the parameters used for random tree generation. For instance, we can favour in the test sets some specific operators (more trigs and exponentials, say), a different distribution of lengths for expressions, and a different balance of constants and variables. We will generate such different test samples with different distributions, and add the results to our paper. These tests should be ready in a few days.

---

> > ### Author Response · Authors · 2020-11-24
> > **Testing on different distributions and generalizability 1/2**
> >
> > Following your suggestion we tested the trained model on datasets with different distributions for operators, variables, constants, expression length and system degrees. The results (given in details below) suggest that changes in the mathematical structure of the systems have little impact on accuracy and generalization. On the other hand, the distribution of the length of sequences input into the transformer impact generalization. The latter could have been expected, as it is a known weakness of transformers, but resilience to change in the mathematical structure is a nice property.
> > Surprisingly, the model is able to generalize to systems that were not in the problem space of the training set. A model trained on systems of 2 to 5 equations manages to achieve high accuracy ($78\%$) on systems of 6 equations, even though it has never seen a system with 6 equations at train time. We detail below the way we constructed the test sets and provide detailed results and interpretations.
> >
> > We modified the data generator to produce new test datasets for end to end stability prediction. Four modifications were considered:
> >
> >  - Unary operators: varying the distribution of operators in the system. In the training data, unary operators are selected at random from a set of nine, three trigonometric functions, three inverse trigonometric functions, logarithm and exponential, and square root (the four basic operations are always present). In this set of experiments, we generated four test sets, without trigonometric functions, without logs and exponentials, only with square roots, and with a different balance of operators (mostly square roots).
> > - Variables and integers: varying the distribution of variables in the system. In the training data, $30\%$ of the leaves are numbers, the rest variables. We changed this probability to $0.1$, $0.5$ and $0.7$. This has no impact on expression length, but higher probabilities make the Jacobians more sparse.
> > - Expression lengths: making expressions longer than in the train set. In the training data, for a system of $n$ equations, we generate functions with $3$ to $2n+3$ operators. In this experiments, we tried functions between $n+3$ and $3n+3$ and $2n+3$ and $4n+3$. This means that the test sequences are, on average, much longer that those seen at training, a known weakness of sequence to sequence models.
> > - Larger degree: our models were trained on systems with $2$ to $5$ equations, we tried to test it on systems with $6$ equations. Again, this usually proves difficult for transformers.
> >
> > Note that the two first sets of experiments feature out-of-distribution tests, exploring different distributions over the same problem space as the training data. The two last sets, on the other hand, explore a different problem space, featuring longer sequences.
> > A table below presents the results of these experiments. Changing the distribution of operators, variables and integers has little impact on accuracy, up to two limiting cases. First, over systems of degree five (the largest in our set, and more difficult for the transformers) change in operator distribution has a small adverse impact on performance (but not change in variable distribution). Second, which the proportion of integers become very large, and therefore Jacobians become very sparse, the degree of the systems has less impact on performance. But overall results remain over $95\%$, and the model proves to be very resistant to changes in distribution over the same problem space.
> >
> > Over systems with longer expressions, overall accuracy tends to decreases. Yet, systems of two or three equations are not affected by a doubling of the number of operators (and sequence length), compared to the training data. Most of the loss in performance concentrates on larger degrees, which suggests that it results from the fact that the transformer is presented at test time with much longer sequences that what it saw at training. In any case, all results but one are well above the fastText baseline ($60.5\%$).
> >
> > When tested on systems with six equations, the trained model predicts stability in $78.7\%$ of cases. This is a very interesting result, where the model is extrapolating out of the problem space (i.e. no system of six equations have been seen during training) with an accuracy well above chance level, and the fastText baseline.

---

> > > ### Author Response · Authors · 2020-11-24
> > > **Testing on different distributions and generalizability 2/2**
> > >
> > > Here is the table of the results
> > >
> > > \begin{array}{l|c|cccc}
> > >         \\hline
> > >                     & \\text{Overall} &  \\text{Degree 2}  & \\text{Degree 3}   & \\text{Degree 4}   & \\text{Degree 5         }\\\\
> > >         \\hline
> > >         \text{Baseline: training distribution}  & 96.4 & 98.4 & 97.3 & 95.9 & 94.1 \\\\
> > >         \\hline
> > >         \\text{Unary operators: no trigs}  & {95.7} & {98.8} & {97.3} & {95.5} & {91.2} \\\\
> > >         \\text{Unary operators: no logs } & {95.3} & {98.2} & {97.1} & {95.2} & {90.8} \\\\
> > >         \\text{Unary operators: no logs and trigs } & {95.7} & {98.8} & {97.7} & {95.2} & {91.0} \\\\
> > >         \\text{Unary operators: less logs and trigs}  & {95.9} & {98.8} & {96.8} & {95.0} & {93.1} \\\\
> > > \\hline
> > >         \\text{Variables and integers: 10\\% integers  }& {96.1} & {98.6} & {97.3} & {94.7} & {93.8} \\\\
> > >         \\text{Variables and integers: 50\\% integers } & {95.6} & {97.8} & {96.7} & {94.3} & {93.1} \\\\
> > >         \\text{Variables and integers: 70\\% integers}  & {95.7} & {95.7} & {95.9} & {95.7} & {95.5} \\\\
> > > \\hline
> > >         \\text{Expression lengths: $n+3$ to $3n+3$ } & {89.5} & {96.5} & {92.6} & {90.0} & {77.9} \\\\
> > >         \\text{Expression lengths: $2n+3$ to $4n+3$ } & {79.3} & {93.3} & {88.3} & {73.4} & {58.2} \\\\
> > > \\hline
> > >         \\text{System degree: degree 6} & 78.7 \\\\
> > > \\hline
> > >     \end{array}

---

### Official Review · AnonReviewer3 · 2020-10-27
**Lack of methodological contribution**

**Rating:** 3
**Confidence:** 4

**Review:**

This paper empirically demonstrated the effectiveness of neural networks for learning to predict different mathematical properties of dynamical systems, which achieve high accuracy on synthetic datasets generated by the authors.


Pros:
+ the paper is clearly written and easy to follow
+ the dataset seems to be carefully generated and is large.

Cons:
- The authors do not clearly state their methodology, including the concrete architecture of the transformer-based model, and the training loss for optimizing the model. As shown in the experiments, the accuracy of the proposed model is much higher than the baseline, but it is not clear what the major reason is, and why.
- As the dataset contains about 50 million samples, and only 10000 of them are held out for test and validation, which means the training dataset contains sufficient data and the result could just be overfitting. Besides, the authors do not describe clearly how they generate the dataset and what is the problem distribution. For some distribution (e.g., those with smaller variance), maybe 50 million is large enough and there won't be a generalization issue. As a well-known fact that neural network has universal approximation ability, it is not surprising that it can learn to predict the mathematical properties given enough data. It will be better if the authors could show a '#training sample VS test accuracy' curve to show how the method behaves differently given different numbers of training samples.

Overall this paper does not have enough technical contributions, so I vote for a clear reject.

---

> ### Author Response · Authors · 2020-11-17
> **Response to Reviewer 3**
>
> Thank you very much for your comments, here are replies to your observations.
>
> ======== The authors do not clearly state their methodology, including the concrete architecture of the transformer-based model, and the training loss for optimizing the model. As shown in the experiments, the accuracy of the proposed model is much higher than the baseline, but it is not clear what the major reason is, and why. ======
>
> Thank you for your remark. Since the architecture we use is identical to the one in ``" Attention is all you need " (Vaswani et al.), we did not describe it in detail, for lack of space. However, we agree that it would make the paper more self-contained, so we will add a section describing the model in the supplementary material.
>
> The training loss we use is the cross entropy between the model predicted output and actual result from the dataset. The model used as a baseline implements a bag of words, which estimates the conditional probability of the (binary) output given the distribution of tokens (or sequences of N tokens) in the input. This model will catch simple correlations between input and output, such as some mathematical operator making systems more unstable, or a correlation between the length of a system and its properties. Because such correlations exist in our problems, our baseline is over chance level. However, bag of words models do not take into account the order of tokens. Transformers handle word order through their attention mechanism, and we believe this explains their better accuracy: the order of tokens in a mathematical expression has an impact on its meaning, and properties.
>
> ====== As the dataset contains about 50 million samples, and only 10000 of them are held out for test and validation, which means the training dataset contains sufficient data and the result could just be overfitting. (...)  As a well-known fact that neural network has universal approximation ability, it is not surprising that it can learn to predict the mathematical properties given enough data. It will be better if the authors could show a '#training sample VS test accuracy' curve to show how the method behaves differently given different numbers of training samples.====
>
> Thank you for this remark. This was clearly not underlined enough in the paper. We agree that neural networks have a universal approximation ability from the universal approximation theorem (UAT). However, the universal approximation theorem states that a neural network of arbitrary depth can approximate any Lebesgue integrable function from a compact subset of the finite-dimensional vector space $R^n$ to $R^n$. What we are trying to approximate here is much more complicated: formally we try to approximate a functional acting of an infinite-dimensional space. As a consequence the UAT does not apply and the underlying reason is really that the space of possible functions is infinite-dimensional and cannot be accurately represented by a finite-dimensional vector space, even large. Therefore, 50 millions examples cannot grasp the generality of all the functions that can be generated and there are an infinite number of functions whose form is not even close to those 50M examples. What we show in this paper, however, is that some mathematical properties like stability or controllability can be learned with so "few" examples (note that most of the models use in fact much less than the 50M examples generated). Overall, this is the reason why we believe that it is not obvious that mathematical properties could be learned from examples and even surprising (at least in such settings). We would also like to emphasize that in general learning mathematical properties may be hard: it was shown in Saxton et al. that neural networks struggle with learning the decomposition in prime numbers. This discussion was probably missing and we will add additional explanations about this in the paper.
>
> Concerning showing a curve "training sample VS test accuracy", we believe this is a very good idea, thank you for the suggestion. We added to the supplementary material a graph, presenting learning curves over twelve runs of the end to end stability experiment.  From one run to the other, datasets are shuffled and the training set is not the same (the test set does not change). Yet, all runs display similar learning curves, showing that accuracy goes up from 55 to 92% over the first five million examples, and saturates over 95% after 10 millions examples. We believe this, together with the size of the problem space estimated in the appendix, rules out overfitting and memorization.
>
> Finally, on a different aspect, we would also like to underline that one interesting point of this work is not only to show that these mathematical properties can be learned, but that they can be learned using language model, i.e. models that were not at all engineered for such purpose, and where the functions and numbers are fed as tokens.

---

> > ### Author Response · Authors · 2020-11-24
> > **Testing on different distributions 1/2**
> >
> > Following your relevant remarks and other reviewers suggestions we tested the trained model on datasets with different distributions for operators, variables, constants, expression length and system degrees. The results (given in details below) suggest that changes in the mathematical structure of the systems have little impact on accuracy and generalization. On the other hand, the distribution of the length of sequences input into the transformer impact generalization. The latter could have been expected, as it is a known weakness of transformers, but resilience to change in the mathematical structure is a nice property.
> > Surprisingly, the model is able to generalize to systems that were not in the problem space of the training set. A model trained on systems of 2 to 5 equations manages to achieve high accuracy ($78\%$) on systems of 6 equations, even though it has never seen a system with 6 equations at train time. We detail below the way we constructed the test sets and provide detailed results and interpretations.
> >
> > We modified the data generator to produce new test datasets for end to end stability prediction. Four modifications were considered:
> >
> >  - Unary operators: varying the distribution of operators in the system. In the training data, unary operators are selected at random from a set of nine, three trigonometric functions, three inverse trigonometric functions, logarithm and exponential, and square root (the four basic operations are always present). In this set of experiments, we generated four test sets, without trigonometric functions, without logs and exponentials, only with square roots, and with a different balance of operators (mostly square roots).
> > - Variables and integers: varying the distribution of variables in the system. In the training data, $30\%$ of the leaves are numbers, the rest variables. We changed this probability to $0.1$, $0.5$ and $0.7$. This has no impact on expression length, but higher probabilities make the Jacobians more sparse.
> > - Expression lengths: making expressions longer than in the train set. In the training data, for a system of $n$ equations, we generate functions with $3$ to $2n+3$ operators. In this experiments, we tried functions between $n+3$ and $3n+3$ and $2n+3$ and $4n+3$. This means that the test sequences are, on average, much longer that those seen at training, a known weakness of sequence to sequence models.
> > - Larger degree: our models were trained on systems with $2$ to $5$ equations, we tried to test it on systems with $6$ equations. Again, this usually proves difficult for transformers.
> >
> > Note that the two first sets of experiments feature out-of-distribution tests, exploring different distributions over the same problem space as the training data. The two last sets, on the other hand, explore a different problem space, featuring longer sequences.
> > A table below presents the results of these experiments. Changing the distribution of operators, variables and integers has little impact on accuracy, up to two limiting cases. First, over systems of degree five (the largest in our set, and more difficult for the transformers) change in operator distribution has a small adverse impact on performance (but not change in variable distribution). Second, which the proportion of integers become very large, and therefore Jacobians become very sparse, the degree of the systems has less impact on performance. But overall results remain over $95\%$, and the model proves to be very resistant to changes in distribution over the same problem space.
> >
> > Over systems with longer expressions, overall accuracy tends to decreases. Yet, systems of two or three equations are not affected by a doubling of the number of operators (and sequence length), compared to the training data. Most of the loss in performance concentrates on larger degrees, which suggests that it results from the fact that the transformer is presented at test time with much longer sequences that what it saw at training. In any case, all results but one are well above the fastText baseline ($60.5\%$).
> >
> > When tested on systems with six equations, the trained model predicts stability in $78.7\%$ of cases. This is a very interesting result, where the model is extrapolating out of the problem space (i.e. no system of six equations have been seen during training) with an accuracy well above chance level, and the fastText baseline.

---

> > > ### Author Response · Authors · 2020-11-24
> > > **Testing on different distributions 2/2**
> > >
> > > Here is the table of the results
> > >
> > > \begin{array}{l|c|cccc}
> > >         \\hline
> > >                     & \\text{Overall} &  \\text{Degree 2}  & \\text{Degree 3}   & \\text{Degree 4}   & \\text{Degree 5         }\\\\
> > >         \\hline
> > >         \text{Baseline: training distribution}  & 96.4 & 98.4 & 97.3 & 95.9 & 94.1 \\\\
> > >         \\hline
> > >         \\text{Unary operators: no trigs}  & {95.7} & {98.8} & {97.3} & {95.5} & {91.2} \\\\
> > >         \\text{Unary operators: no logs } & {95.3} & {98.2} & {97.1} & {95.2} & {90.8} \\\\
> > >         \\text{Unary operators: no logs and trigs } & {95.7} & {98.8} & {97.7} & {95.2} & {91.0} \\\\
> > >         \\text{Unary operators: less logs and trigs}  & {95.9} & {98.8} & {96.8} & {95.0} & {93.1} \\\\
> > > \\hline
> > >         \\text{Variables and integers: 10\\% integers  }& {96.1} & {98.6} & {97.3} & {94.7} & {93.8} \\\\
> > >         \\text{Variables and integers: 50\\% integers } & {95.6} & {97.8} & {96.7} & {94.3} & {93.1} \\\\
> > >         \\text{Variables and integers: 70\\% integers}  & {95.7} & {95.7} & {95.9} & {95.7} & {95.5} \\\\
> > > \\hline
> > >         \\text{Expression lengths: $n+3$ to $3n+3$ } & {89.5} & {96.5} & {92.6} & {90.0} & {77.9} \\\\
> > >         \\text{Expression lengths: $2n+3$ to $4n+3$ } & {79.3} & {93.3} & {88.3} & {73.4} & {58.2} \\\\
> > > \\hline
> > >         \\text{System degree: degree 6} & 78.7 \\\\
> > > \\hline
> > >     \end{array}

---

### Official Review · AnonReviewer2 · 2020-10-29
**Really neat new dataset and application**

**Rating:** 7
**Confidence:** 4

**Review:**

=Quality=
High: well executed and motivated

=Clarity=
Well-situated wrt to related work.  Baseline needs to be explained more (see below).

=Originality=
The ML method is standard. The novelty is in setting up the datasets and evaluation metrics. Doing was technically complex. The datasets very valuable and will hopefully be open-sourced.

=Significance=
This paper demonstrates that neural networks are surpisingly good at the task of predicting certain properties of differential systems, such as their stability. This is a neat result and is sufficient for publication. However, the paper would have more impact if there was a concrete proof of concept of how such a tool could be used to improve the lives of practioners.

=Motivation/Introduction=
I agree with you that it is very impressive that neural networks can nearly solve these tasks. It would be helpful if you devoted more of the exposition to explaining how your classifiers could be used by practitioners to improve their workflows. Are there engineering applications, for example,  where the classifier could be used to quickly screen proposed systems?

Also, I'd like to better understand what the novel capability of your tool is. Is it faster than numerical methods for verifying these properties? More accurate? Applicable in situations where symbolic manipulation is impossible?

=Predicting Control feedback matrices=
I found this section very cool and encouraging for future work! I'd mention it in the intro more.


=Baseline classifier=
My primary hesitation with the paper is that there is not enough justification for why your baseline is sufficient.

You need to provide far more details about the FastText baseline. Describing the name of the software package is insufficient. What is the actual model?

"Such high performances over difficult mathematical tasks may come as a surprise, and one
might wonder whether the model is exploiting some defect in the dataset, or some trivial
property of the problems that would allow an easy way to correct solutions. We believe
this is very unlikely...because a trivial solution would be found by the text classification tool we use as a baseline."

This argument is weak without an explanation of what the baseline model is and what its inductive biases are.


Are there no heuristics from the application community that would serve as baselines?


In the discussion, you mention in passing that "providing at train time intermediate results that would help a human calculator (frequencies for PDE, or Jacobians for stability) does not improve accuracy." It seems to me that a baseline method would be a simple machine learning model on top of some hand-crafted features of these intermediate results.


=Discussion section=
"in some of our problems, even a model with one layer and 64 dimensions
obtains a high accuracy, and such a small model would never be able to memorize that
many examples."
Either make this precise or remove it. At first glance, it seems to be that 64 dimensions has a lot of capacity. For example, 2^64 is much bigger than the size of your dataset.


=Open Sourcing=
Will you be able to open-source the datasets? Doing so would considerably increase the future impact of your work, as it would provide a benchmark for future ML methods.

=Evaluation set=
It would be cool if you could have a couple of anecdotes of applying your classifier to famous equations from papers, particularly ones where the derivations to prove stability, for example, were quite tedious. You argue that the test set is representative, since it is uniformly sampled, but are the equations of interest to the community in some corner of this space?

=Analysis/interpretation=

Do the attention patterns of the transformer reveal anything interesting?

What seems to characterize the equations that the model makes mistakes on?

Can you use your model to get  per-equation embeddings? Do they reveal interesting cluster structure in the data?

*** After reading the authors' responses ***
I have raised my score to a 7. I felt that some of the key questions, e.g. regarding generalization to mathematical expressions that are qualitatively different than the training data,  were answered well. This paper should not be reviewed as being methods-driven. It's about demonstrating a new way that deep learning could be transformative for engineering, by allowing engineers to screen proposed designs for stability, etc.

---

> ### Author Response · Authors · 2020-11-17
> **Response to reviewer 2 - 1/4**
>
> Thanks you very much for your review, here is a preliminary reply (the last part, about potential applications will be updated later)
>
>  ===== I found this section very cool and encouraging for future work! I'd mention it in the intro more. ====
>
> Thanks for finding this cool! It is indeed encouraging for future work that could try to predict similarly other mathematical/numerical quantities. We'll emphasize this in the intro following your comment.
>
> ==== My primary hesitation with the paper is that there is not enough justification for why your baseline is sufficient.
> You need to provide far more details about the FastText baseline. Describing the name of the software package is insufficient. What is the actual model? ========
>
> Defining a baseline for such tasks is difficult. Since all the problems we study can be solved by mathematical libraries, a "math library" baseline would be 100%. On the other hand, there are no previous attempts to solve these problems with deep learning models. We retained fastText because it is a basic but state-of-the-art NLP model, which has proven to be as powerful as deep learning classifiers.
>
> We will add a short presentation of fastText in the updated version of the paper. It implements a bag of words, which estimates the conditional probability of a binary output given the distribution of tokens (or fixed sequences up to N tokens, here N=5) in the input. Bag of words are often used for sentiment analysis (e.g. detecting whether a comment is positive or negative), and will pick up simple correlations between inputs and outputs (like stability being dependent on the degree of the system, or the presence of some specific operator). Another virtue of fastText is that it would pick up obvious solutions due to a trivial case of the problem, or a glitch in the generating procedure. In this respect, we believe it provides a relevant baseline, as a minimal NLP solution for these problems.
>
> ===== Are there no heuristics from the application community that would serve as baselines? =====
>
> Heuristics from the application community are an interesting idea, and there are indeed a few that could serve as baselines. Unfortunately, they are limited to simple or trivial cases. For the controllability problem, for instance, the answer can be guessed if: one variable is missing in all the right-hand sides of the system; an equation only depends on variables that are themselves uncontrolled; the system is triangular with non-zero dominant coefficients, etc. But these special cases will nearly never appear in randomly generated systems, and the baseline would be very low. Besides, most of those cases are likely to be picked up by fastText. Such heuristics could be used for testing, though, by using a dataset of special cases and assessing the ability of the model on these cases. It is all the more interesting as such cases will almost never appear in the training set.
>
> ===== Either make this precise or remove it. At first glance, it seems to be that 64 dimensions has a lot of capacity. For example, 2^64 is much bigger than the size of your dataset. =====
>
> You are fully correct in raising this point, thank you for the remark. We will change this sentence. Our idea is that under the constraints of gradient learning, it would probably take a lot more than 64 dimensions to memorize 50 million examples.
> What we meant was that, in regular NLP, such shallow and low dimension transformers are too small to learn even the basic syntax of natural languages.
>
> ===== Will you be able to open-source the datasets? Doing so would considerably increase the future impact of your work, as it would provide a benchmark for future ML methods. =====
>
> Yes, of course. We will not only open-source the dataset but also the code of the generators for each task so that it should be fairly easy to modify it should someone want to apply this approach to another application.
>
> ====== It would be cool if you could have a couple of anecdotes of applying your classifier to famous equations from papers, particularly ones where the derivations to prove stability, for example, were quite tedious. You argue that the test set is representative, since it is uniformly sampled, but are the equations of interest to the community in some corner of this space? =====
>
> Thank you for this nice suggestion. That would be cool indeed, and the equations of interest to the community are definitely in our space of function. We will add a subsection in Appendix with the outputs of our model when applied on famous/ textbook equations: Schroedinger, heat equations for the PDEs, damped/amplified transport equations ; an example of stability analysis taken from a maths paper where the proof of stability is tedious and requires some change of variables (Bando-FtL model for traffic flow) ; some example of controllability taken from a classic textbook (inverted Cart-Pendulum). They might not be ready for the rebuttal, but they will for the final version.

---

> > ### Author Response · Authors · 2020-11-17
> > **Response to reviewer 2 - 2/4**
> >
> > ===== Do the attention patterns of the transformer reveal anything interesting?
> > What seems to characterize the equations that the model makes mistakes on?
> > Can you use your model to get per-equation embeddings? Do they reveal interesting cluster structure in the data? =====
> >
> > We tried to analyze the model behavior by looking at the attention heads of the model, and the tokens the models focus on when it predicts a specific sequence, following the paper ``What does bert look at? an analysis of bert's attention'' by Clark et al., 2019. Unfortunately, we were not able to extract specific patterns from this analysis. Unlike what was observed in NLP, we found that each head in the model tends to attends much more tokens than in natural language (i.e. the attention weights are quite uniformly distributed), and this from the first layer. As a result, the information about the input tokens are spread quite early in the layers of the model, which makes interpretation very difficult. We tried to reduce the number of hidden states a head can attend by using a top-k on the attention weights, to see if this facilitates the visualization. Unfortunately, this modification in the model deteriorated the performance quite significantly, and we did not investigate more in this direction.
> >
> > We ran a sequence-to-sequence model without attention, so that each input equation is mapped to a fixed-sized representation. We then fed a set of input equations into the model, and used a t-SNE visualization to see whether we can observe clusters of equations. What we observed is mainly that equations with nearby representations have similar length / tokens. However, even embeddings in similar locations can lead to different decoded sequences. Potentially, based on how the computation is split between the encoder and the decoder, the decoder could be doing the majority of the work, which means that the representations provided by the encoder would not be so meaningful. If the computation is mainly done by the encoder, this is not something we managed to conclude from the visualizations.

---

> > > ### Author Response · Authors · 2020-11-18
> > > **Response to Reviewer 2 - 3/4**
> > >
> > > ======Motivation/Introduction= I agree with you that it is very impressive that neural networks can nearly solve these tasks. It would be helpful if you devoted more of the exposition to explaining how your classifiers could be used by practitioners to improve their workflows. Are there engineering applications, for example, where the classifier could be used to quickly screen proposed systems?
> > >
> > > Also, I'd like to better understand what the novel capability of your tool is. Is it faster than numerical methods for verifying these properties? More accurate? Applicable in situations where symbolic manipulation is impossible?========
> > >
> > > Thank you for this suggestion, it would indeed be interesting to compare the pros and cons of using our tools vs. the usual numerical algorithms. As we are primarily trying to show that a language model can be used to learn well-known mathematical properties of the classical theory, the symbolic manipulation and resolution of the problems we consider is always possible. However for practical use, our tool seems to be much faster than the current numerical methods, especially for the non-autonomous controllability; the prediction of stabilizing feedback matrices. This can be summarized in the following table giving the time (in seconds) needed to solve the different problems per system. The numerical algorithms run in python and uses libraries of numpy and/or scipy and/or sympy depending on the task.
> > > \begin{array} {|l|c|c|}\\hline \text{Tasks} & \text{Python} & \text{Transformer} \\\\ \\hline \text{Stability end-to-end} & 0.02 & 0.0008 \\\\  \text{Stability largest eigenvalue} & 0.02 & 0.002 \\\\ \\hline \text{Controllability (autonomous)} & 0.05 & 0.001 \\\\ \\hline \text{Predicting a feedback matrix} & 0.4 & 0.002 \\\\  \\hline \end{array}
> > > The numerical algorithms in python might have room for improvement, even though they use classical libraries, but there is a speed-up factor 10 to 200 when using the transformer which suggests that transformers outperform by far the use of classical numerical algorithms. We remark that the speed of the transformer is about the same for all the tasks while the speed of the classical numerical algorithms changes a lot. One explanation would be that a trained network with a given architecture will have about the same speed whatever the task for inputs of the same size, while the classical methods to solve the mathematical problems depends clearly on the difficulty of the task for a fixed size of input. This is another argument to use transformers for solvable but computationally costly mathematical problems. We will include in Appendix some of these comparisons.
> > > Concerning the prediction of stabilizing feedback matrices our tool has another interesting advantage: most of the time it predicts different feedback matrices than the one that would be usually computed with the classical methods. Of course there would probably be a theoretical way to derive them, but they are not given directly by the usual formula used by the classical numerical algorithms, therefore our tool might introduce some more diversity.
> > >
> > >
> > > On a higher level, this approach could interest scientists of the applicative community in two other ways:
> > > - There are stability-like problems for which one has no classical numerical method that would work in any cases, but many methods that would each work in some particular cases. What could be done would be to use these methods separately to create a large dataset as we are doing here, and then learn to solve the problem with our model. We would then have a unique tool that would replace all the particular methods. To be fair this idea does not come from us: we have been recently contacted by computational biologists who would like to work with us on such a tool after seeing this paper.
> > > - Here the problems we study have known solutions, but the high accuracy results suggest that the same approach could be tried on open or computationally hard problems. The only requirements are building a training sample (either by solving some representative cases or generating problems from known solutions), and being able to check the validity of solutions (which is usually much easier than solving the problem). Then, a trained model could guess solutions of open problems, which would be interesting and very useful in mathematics even with low accuracy.
> > >
> > > Finally, in the long term, it is likely that scientists will try to automatically derive mathematical or physical models using machine learning. Our research could provide a way to screen those models that have some relevant property, or select the most promising models in a large generated batch, so that scientist can focus on improving them.

---

> > > > ### Author Response · Authors · 2020-11-18
> > > > **Response to Reviewer 2 - 4/4**
> > > >
> > > > ===========In the discussion, you mention in passing that "providing at train time intermediate results that would help a human calculator (frequencies for PDE, or Jacobians for stability) does not improve accuracy." It seems to me that a baseline method would be a simple machine learning model on top of some hand-crafted features of these intermediate results.==========
> > > >
> > > > Thank you for this suggestion. We could definitely try to feed more systematically as input some intermediate results (either steps that would be useful to a human, or steps used in the classical theory) and see whether these features improve the performance of the model. This should give us some insights about what the model is doing. Indeed, if adding intermediary results from the classical theory improves the performance significantly, it would suggest that the model is using a method different from the theory, otherwise these additional features would be redundant and would not improve the performance of the model. We will try to see if we can include this in time for the rebuttal, otherwise we will include it in a future version of the paper.

---

> > > > > ### Author Response · Authors · 2020-11-24
> > > > > **Testing on different distributions 1/2**
> > > > >
> > > > > Following your suggestion we tested the trained model on datasets with different distributions for operators, variables, constants, expression length and system degrees. The results (given in details below) suggest that changes in the mathematical structure of the systems have little impact on accuracy and generalization. On the other hand, the distribution of the length of sequences input into the transformer impact generalization. The latter could have been expected, as it is a known weakness of transformers, but resilience to change in the mathematical structure is a nice property.
> > > > > Surprisingly, the model is able to generalize to systems that were not in the problem space of the training set. A model trained on systems of 2 to 5 equations manages to achieve high accuracy ($78\%$) on systems of 6 equations, even though it has never seen a system with 6 equations at train time. We detail below the way we constructed the test sets and provide detailed results and interpretations.
> > > > >
> > > > > We modified the data generator to produce new test datasets for end to end stability prediction. Four modifications were considered:
> > > > >
> > > > >  - Unary operators: varying the distribution of operators in the system. In the training data, unary operators are selected at random from a set of nine, three trigonometric functions, three inverse trigonometric functions, logarithm and exponential, and square root (the four basic operations are always present). In this set of experiments, we generated four test sets, without trigonometric functions, without logs and exponentials, only with square roots, and with a different balance of operators (mostly square roots).
> > > > > - Variables and integers: varying the distribution of variables in the system. In the training data, $30\%$ of the leaves are numbers, the rest variables. We changed this probability to $0.1$, $0.5$ and $0.7$. This has no impact on expression length, but higher probabilities make the Jacobians more sparse.
> > > > > - Expression lengths: making expressions longer than in the train set. In the training data, for a system of $n$ equations, we generate functions with $3$ to $2n+3$ operators. In this experiments, we tried functions between $n+3$ and $3n+3$ and $2n+3$ and $4n+3$. This means that the test sequences are, on average, much longer that those seen at training, a known weakness of sequence to sequence models.
> > > > > - Larger degree: our models were trained on systems with $2$ to $5$ equations, we tried to test it on systems with $6$ equations. Again, this usually proves difficult for transformers.
> > > > >
> > > > > Note that the two first sets of experiments feature out-of-distribution tests, exploring different distributions over the same problem space as the training data. The two last sets, on the other hand, explore a different problem space, featuring longer sequences.
> > > > > A table below presents the results of these experiments. Changing the distribution of operators, variables and integers has little impact on accuracy, up to two limiting cases. First, over systems of degree five (the largest in our set, and more difficult for the transformers) change in operator distribution has a small adverse impact on performance (but not change in variable distribution). Second, which the proportion of integers become very large, and therefore Jacobians become very sparse, the degree of the systems has less impact on performance. But overall results remain over $95\%$, and the model proves to be very resistant to changes in distribution over the same problem space.
> > > > >
> > > > > Over systems with longer expressions, overall accuracy tends to decreases. Yet, systems of two or three equations are not affected by a doubling of the number of operators (and sequence length), compared to the training data. Most of the loss in performance concentrates on larger degrees, which suggests that it results from the fact that the transformer is presented at test time with much longer sequences that what it saw at training. In any case, all results but one are well above the fastText baseline ($60.5\%$).
> > > > >
> > > > > When tested on systems with six equations, the trained model predicts stability in $78.7\%$ of cases. This is a very interesting result, where the model is extrapolating out of the problem space (i.e. no system of six equations have been seen during training) with an accuracy well above chance level, and the fastText baseline.

---

> > > > > > ### Author Response · Authors · 2020-11-24
> > > > > > **Testing on different distributions 2/2**
> > > > > >
> > > > > > Here is the table of the results
> > > > > >
> > > > > > \begin{array}{l|c|cccc}
> > > > > >         \\hline
> > > > > >                     & \\text{Overall} &  \\text{Degree 2}  & \\text{Degree 3}   & \\text{Degree 4}   & \\text{Degree 5         }\\\\
> > > > > >         \\hline
> > > > > >         \text{Baseline: training distribution}  & 96.4 & 98.4 & 97.3 & 95.9 & 94.1 \\\\
> > > > > >         \\hline
> > > > > >         \\text{Unary operators: no trigs}  & {95.7} & {98.8} & {97.3} & {95.5} & {91.2} \\\\
> > > > > >         \\text{Unary operators: no logs } & {95.3} & {98.2} & {97.1} & {95.2} & {90.8} \\\\
> > > > > >         \\text{Unary operators: no logs and trigs } & {95.7} & {98.8} & {97.7} & {95.2} & {91.0} \\\\
> > > > > >         \\text{Unary operators: less logs and trigs}  & {95.9} & {98.8} & {96.8} & {95.0} & {93.1} \\\\
> > > > > > \\hline
> > > > > >         \\text{Variables and integers: 10\\% integers  }& {96.1} & {98.6} & {97.3} & {94.7} & {93.8} \\\\
> > > > > >         \\text{Variables and integers: 50\\% integers } & {95.6} & {97.8} & {96.7} & {94.3} & {93.1} \\\\
> > > > > >         \\text{Variables and integers: 70\\% integers}  & {95.7} & {95.7} & {95.9} & {95.7} & {95.5} \\\\
> > > > > > \\hline
> > > > > >         \\text{Expression lengths: $n+3$ to $3n+3$ } & {89.5} & {96.5} & {92.6} & {90.0} & {77.9} \\\\
> > > > > >         \\text{Expression lengths: $2n+3$ to $4n+3$ } & {79.3} & {93.3} & {88.3} & {73.4} & {58.2} \\\\
> > > > > > \\hline
> > > > > >         \\text{System degree: degree 6} & 78.7 \\\\
> > > > > > \\hline
> > > > > >     \end{array}

---

### Official Review · AnonReviewer4 · 2020-10-31
**Clear, well-executed, interesting paper**

**Rating:** 8
**Confidence:** 4

**Review:**

This paper shows that transformer models can be used to accurately learn advanced mathematical computations from millions of examples.  The problems are drawn from the fields of differential equations and control theory.  The selected problems are ones that are solvable using known algorithms; however, these algorithms involve a sequence of advanced mathematical operations (e.g., differentiation, calculating the rank of a matrix, calculating the eigenvalues of a matrix, etc), for which no known simple shortcuts exist. For the experiments in this paper, for each problem a large number (50 million) of training examples are randomly generated, and are then used to train a transformer model.  Across these problems, the paper shows that the neural network is able to solve these problems at high accuracy (96-99.7% accuracy).

Strengths
- The paper ask a well-motivated question regarding whether neural networks are able to learn complex mathematical operations from examples.
- The paper is clearly written, and the experimental rigor/quality appears quite high.
- The empirical results are quite intriguing, and raises many interesting questions for future research.  For example, (1) how is a transformer model managing to attain such high accuracy on these problems, (2) what other complicated mathematical problems might be similarly learnable, (3) what are the practical implications to real systems of these results.
- The paper does a good job considering the various potential counterarguments to its conclusions in the discussion section (Section 5.4).  For example, the paper argues convincingly that (1) it’s unlikely the model is exploiting some trivial distinction between the positive vs negative examples because the examples are sampled randomly, and that (2) it’s unlikely the model is interpolating between solutions because the problem space is so much larger than the training set (and because small models which would be unable to memorize the training set also perform well).

 Weaknesses
- I would have appreciated more of a discussion about the computational cost of solving these problems mathematically vs. solving them with a neural network.  What is the computational complexity (Big-O) of each of the known mathematical algorithms for solving these problems?  Are there large computational savings from using a neural network?  What are the practical implications of the results shown in this paper?
- I think including a few more baselines would have been useful.  Also, a brief description of the FastText model would make the paper more self-contained.   One question: Why is the FastText model in Section 5.1 only trained with 2 million examples, while the transformer model is trained with 50 million examples?
- In the current experiments, the test set is drawn from the same exact (random) distribution as the training set.  I was very curious whether the model would have been able to attain high test-time accuracy, had the test examples been drawn from a different distribution.  In particular, I’d be curious how difficult it would be to construct a test distribution on which the current model performs terribly.  This line of questioning would be able to start better answering whether the model is solving the problem in a way that truly generalizes.
- I think the paper could be strengthened by adding additional error analysis to try to better understand the errors made by the transformer model.
- The paper says “training is performed on 8 V100 GPUs with float16 operations”: It would have been nice to hear more about the training process: For example, low long did training take for the various problems, and how was training distributed across the GPUs?
- I would have appreciated more of a discussion about the broader implications and significance of these results.

Overall, I thought the paper was very interesting and well executed, and I think it would make a very nice addition to ICLR 2021.

---

> ### Author Response · Authors · 2020-11-17
> **Response to Reviewer 4 - 2/2**
>
> ===== The paper says “training is performed on 8 V100 GPUs with float16 operations”: It would have been nice to hear more about the training process: For example, low long did training take for the various problems, and how was training distributed across the GPUs? ====
>
> On autonomous control, it took 11 hours for the model to reach its best accuracy of 97.4%. However, the performance was already of 95% after 4 hours. Numerical computations took more time. For the stability experiment, our best model reached its best accuracy of 86.8% after 76 hours of training, while the accuracy was only of 80% after 24 hours.  These figures are typical, on average, qualitative tasks took about 12 hours to train, and computations between two and three days. We added a comment in the paper to this effect.
>
> The model is distributed across GPUs, so that all GPUs have access to the same shared copy of the model. At each iteration, each GPU processes an independently generated batch, and updates the model weights based on the gradients averaged over all workers. Overall, this setup is equivalent to training on a single GPU, but with N times larger batches (where N is the number of GPUs). We will provide additional details on the training process in the updated version of the paper.
>
> ===== I would have appreciated more of a discussion about the broader implications and significance of these results. =====
>
> Our results expand on Lample and Charton (2020) findings that transformers can be trained to perform symbolic mathematics. One practical limitation of their results was the fact that many scientific problems include a mixture of symbolic and numerical computations. And prior work on arithmetic had suggested that neural networks did not perform very well on pure computational tasks. Our research shows that transformers can handle mixed problems that involve both symbolic and numerical operations. This suggests that the potential use of transformers in science might be larger than previously thought and be of large interest in many areas where this framework with both symbolic and numerical operations appear: physics, computational-biology, theoretical chemistry, etc.
>
> That transformers can learn complex computations from examples is highly non-intuitive. Especially, providing some intermediate results at train time as additional inputs does not seem to improve accuracy, which suggests that transformers exploit shortcuts to solve these problems. Given the accuracy, these shortcuts are likely to be another way of solving the problem which is mathematically equivalent to the classical way. As such equivalent way is not yet known, it would be very interesting to understand how these shortcut works.
>
> Also, the problems selected here have known solutions, but the same approach could be tried on open or computationally hard problems. The only requirements are building a training sample (either by solving some representative cases or generating problems from known solutions), and being able to check the validity of solutions (which is usually much easier than solving the problem). If this can be done, a trained model could guess solutions of open problems, which would be interesting and very useful in mathematics even if the accuracy is low.
>
> In the long term, it is likely that scientists will try to automatically derive mathematical or physical models using machine learning. Our research could provide a way to screen those models that have some relevant property, or select the most promising models in a large generated batch, so that scientist can focus on improving them.

---

> > ### Author Response · Authors · 2020-11-24
> > **Testing the models on different distributions 1/2**
> >
> > Following your suggestion we tested the trained model on datasets with different distributions for operators, variables, constants, expression length and system degrees. The results (given in details below) suggest that changes in the mathematical structure of the systems have little impact on accuracy and generalization. On the other hand, the distribution of the length of sequences input into the transformer impact generalization. The latter could have been expected, as it is a known weakness of transformers, but resilience to change in the mathematical structure is a nice property.
> > Surprisingly, the model is able to generalize to systems that were not in the problem space of the training set. A model trained on systems of 2 to 5 equations manages to achieve high accuracy ($78\%$) on systems of 6 equations, even though it has never seen a system with 6 equations at train time. We detail below the way we constructed the test sets and provide detailed results and interpretations.
> >
> > We modified the data generator to produce new test datasets for end to end stability prediction. Four modifications were considered:
> >
> >  - Unary operators: varying the distribution of operators in the system. In the training data, unary operators are selected at random from a set of nine, three trigonometric functions, three inverse trigonometric functions, logarithm and exponential, and square root (the four basic operations are always present). In this set of experiments, we generated four test sets, without trigonometric functions, without logs and exponentials, only with square roots, and with a different balance of operators (mostly square roots).
> > - Variables and integers: varying the distribution of variables in the system. In the training data, $30\%$ of the leaves are numbers, the rest variables. We changed this probability to $0.1$, $0.5$ and $0.7$. This has no impact on expression length, but higher probabilities make the Jacobians more sparse.
> > - Expression lengths: making expressions longer than in the train set. In the training data, for a system of $n$ equations, we generate functions with $3$ to $2n+3$ operators. In this experiments, we tried functions between $n+3$ and $3n+3$ and $2n+3$ and $4n+3$. This means that the test sequences are, on average, much longer that those seen at training, a known weakness of sequence to sequence models.
> > - Larger degree: our models were trained on systems with $2$ to $5$ equations, we tried to test it on systems with $6$ equations. Again, this usually proves difficult for transformers.
> >
> > Note that the two first sets of experiments feature out-of-distribution tests, exploring different distributions over the same problem space as the training data. The two last sets, on the other hand, explore a different problem space, featuring longer sequences.
> > A table below presents the results of these experiments. Changing the distribution of operators, variables and integers has little impact on accuracy, up to two limiting cases. First, over systems of degree five (the largest in our set, and more difficult for the transformers) change in operator distribution has a small adverse impact on performance (but not change in variable distribution). Second, which the proportion of integers become very large, and therefore Jacobians become very sparse, the degree of the systems has less impact on performance. But overall results remain over $95\%$, and the model proves to be very resistant to changes in distribution over the same problem space.
> >
> > Over systems with longer expressions, overall accuracy tends to decreases. Yet, systems of two or three equations are not affected by a doubling of the number of operators (and sequence length), compared to the training data. Most of the loss in performance concentrates on larger degrees, which suggests that it results from the fact that the transformer is presented at test time with much longer sequences that what it saw at training. In any case, all results but one are well above the fastText baseline ($60.5\%$).
> >
> > When tested on systems with six equations, the trained model predicts stability in $78.7\%$ of cases. This is a very interesting result, where the model is extrapolating out of the problem space (i.e. no system of six equations have been seen during training) with an accuracy well above chance level, and the fastText baseline.

---

> > > ### Author Response · Authors · 2020-11-24
> > > **Testing on different distributions 2/2**
> > >
> > > Here is the table of the results
> > >
> > > \begin{array}{l|c|cccc}
> > >         \\hline
> > >                     & \\text{Overall} &  \\text{Degree 2}  & \\text{Degree 3}   & \\text{Degree 4}   & \\text{Degree 5         }\\\\
> > >         \\hline
> > >         \text{Baseline: training distribution}  & 96.4 & 98.4 & 97.3 & 95.9 & 94.1 \\\\
> > >         \\hline
> > >         \\text{Unary operators: no trigs}  & {95.7} & {98.8} & {97.3} & {95.5} & {91.2} \\\\
> > >         \\text{Unary operators: no logs } & {95.3} & {98.2} & {97.1} & {95.2} & {90.8} \\\\
> > >         \\text{Unary operators: no logs and trigs } & {95.7} & {98.8} & {97.7} & {95.2} & {91.0} \\\\
> > >         \\text{Unary operators: less logs and trigs}  & {95.9} & {98.8} & {96.8} & {95.0} & {93.1} \\\\
> > > \\hline
> > >         \\text{Variables and integers: 10\\% integers  }& {96.1} & {98.6} & {97.3} & {94.7} & {93.8} \\\\
> > >         \\text{Variables and integers: 50\\% integers } & {95.6} & {97.8} & {96.7} & {94.3} & {93.1} \\\\
> > >         \\text{Variables and integers: 70\\% integers}  & {95.7} & {95.7} & {95.9} & {95.7} & {95.5} \\\\
> > > \\hline
> > >         \\text{Expression lengths: $n+3$ to $3n+3$ } & {89.5} & {96.5} & {92.6} & {90.0} & {77.9} \\\\
> > >         \\text{Expression lengths: $2n+3$ to $4n+3$ } & {79.3} & {93.3} & {88.3} & {73.4} & {58.2} \\\\
> > > \\hline
> > >         \\text{System degree: degree 6} & 78.7 \\\\
> > > \\hline
> > >     \end{array}

---

> ### Author Response · Authors · 2020-11-17
> **Response to Reviewer 4 - 1/2**
>
> Thank you very much for your comments, here is a preliminary reply.
>
> ===== I would have appreciated more of a discussion about the computational cost of solving these problems mathematically vs. solving them with a neural network. What is the computational complexity (Big-O) of each of the known mathematical algorithms for solving these problems? Are there large computational savings from using a neural network? What are the practical implications of the results shown in this paper? ====
>
> Looking at the complexity of numerical algorithms is indeed a natural idea and a good suggestion. Let n be the system size, p the number of variables and q the average length (in tokens) of functions in the system. In all problems considered here, we have $p = O(n)$. Differentiating or evaluating an expression with q tokens is $O(q)$, and calculating the Jacobian of our system is $O(npq)$, i.e. $O(n^2 q)$.
>
> In the stability experiment, calculating the eigenvalues of the Jacobian will be $O(n^3)$ in most practical situations. In the autonomous controllability experiments, construction of the $n \times np$ Kalman matrix is $O(n^3 p)$, and computing its rank, via singular value decomposition or any equivalent algorithm, will be $O(n^3 p)$ as well. The same complexity arise for feedback matrix computations (multiplication, exponentiation and inversion are all $O(n^3)$ for a square n matrix). As a result, for controllability, complexity is $O(n^4)$. Overall, the classical algorithms have a complexity of $O(n^2 q)$ for Jacobian calculation, and $O(n^3)$ (stability) and $O(n^4)$ (controllability) for the problem specific computations.
>
> Current transformer architectures are quadratic in the length of the sequence, in our case $nq$, so a transformer will be $O(n^2 q^2)$ (in speed and memory usage). Therefore, the final comparison will depend on how q, the average length of equations, varies with $n$, the number of parameters. If $q=O(1)$ or $O(log(n))$, transformers have a large advantage over classical methods. This means sparse Jacobians, a condition often met in practice. For controllability, the advantage remains if $q=O(n^{1/2})$, and the two methods are asymptotically equivalent if $q=O(n)$.
>
> However, current research is working on improving transformer complexity to log-linear or linear. If this happened (and there seem to be no theoretical reason preventing it), transformers would have lower asymptotic complexity in all cases.
>
> For given values of n and q, transformers might run faster than most mathematical algorithms, because they use simple matrix operations that can easily be parallelized.
>
> ===== I think including a few more baselines would have been useful. Also, a brief description of the FastText model would make the paper more self-contained. One question: Why is the FastText model in Section 5.1 only trained with 2 million examples, while the transformer model is trained with 50 million examples? ======
>
> Baselines for such problems are difficult to define. From a mathematical standpoint, every problem can be solved using mathematics software, so baseline accuracy would be $100%$. From a deep learning standpoint, there is no pre-existing state of the art. We considered FastText because of its simplicity/efficiency trade-off. Although it is based on bag of words, fastText is sometimes on par with deep learning classifiers in terms of accuracy (sometimes reaching the performance of 29-layer deep models when it uses N-gram features). Its main virtue is that it rules out obvious solutions, due to the specifics of one problem, or glitches in the data generator. We will add a brief presentation of fastText in the paper, thank you for your suggestion.
>
> re: fastText trained over 2 million examples only. Being simpler models, bag of words need less data than sequence-to-sequence models to train to saturation. On the data from section 5.1, fastText accuracy was 59.7% after 500 000 examples, 60.3% after 1 million, 60.4% after 1.5 million, 60.5% after 1.75 million examples, and 60.6% after 2 million examples, and did not increase after that. Over the same dataset, transformers needed 22 millions examples to achieve their best solution, but their accuracy tended to saturate after 5-10 million examples.
>
> ======== In the current experiments, the test set is drawn from the same exact (random) distribution as the training set. I was very curious whether the model would have been able to attain high test-time accuracy, had the test examples been drawn from a different distribution. (...) =====
>
> Thank you for this suggestion! It would be interesting to see this indeed. We will generate different distributions for test sets on the stability problem, by varying the parameters used for random tree generation. For instance, we can bias the test set toward specific operators, a different distribution of expression lengths or a different balance between constants and variables. This analysis should be ready in a few days.

---

> ### Author Response · Authors · 2020-11-18
> **Speed comparison between a trained transformer and math libraries**
>
> As a complement to our reply on your first point (asymptotic speed comparisons between our methods and known mathematical algorithms), we ran a series of experiments, comparing the speed of our trained models to the mathematical algorithms implemented in Python (using Numpy, Sympy, Scipy), over 10000 random problems. Below are the average times (in seconds) needed to solve one problem.
>
> \begin{array} {|l|c|c|} \\hline \text{Tasks} & \text{Python} & \text{Transformer} \\\\
> \\hline \text{Stability end-to-end} & 0.02 & 0.0008 \\\\
> \\hline \text{Stability largest eigenvalue} & 0.02 & 0.002 \\\\
> \\hline \text{Controllability (autonomous)} & 0.05 & 0.001 \\\\
> \\hline \text{Predicting a feedback matrix} & 0.4 & 0.002 \\\\
> \\hline \end{array}
>
> Overall, trained transformers accelerate the calculation by a factor 10 to 200, depending on the problem. It should be noted that transformers operate at the same speed for all problems. This is because computation speed over a trained neural network, is a function of the size and architecture of the network, and not of the nature of the problem solved. The speed of mathematical algorithms, on the other hand, is very dependent on the number of steps they include, and the presence of specific, costly, operations (e.g. integration when computing feedback matrices)

---

### Decision · Program_Chairs · 2021-01-07
**Final Decision**

**Decision:**

Accept (Poster)

**Comment:**

This paper shows that transformer models can be used to learn certain advanced mathematical concepts such as the local stability of differential equations. Reviewers found this surprising and useful for engineers, and the evaluation was adequate. They also felt that it opens the doors to similar studies on other aspects of mathematics.